# A Descriptor-Based Multi-Cluster Memory for Test-Time Adaptation

## Abstract

Test-time adaptation (TTA) aims to preserve model robustness under distribution shifts without access to source data. However, existing memory designs, often based on single clusters or naive sample storage, struggle to capture the diversity of target distributions and adapt efficiently over time. We introduce Multi-Cluster Memory (MCM), a novel memory management framework that organizes samples into multiple clusters using lightweight statistical descriptors such as sample means and variances. The inter-cluster distance naturally expands the coverage of the sample distribution, supports on-demand cluster creation for novel patterns, and maintains bounded capacity through an Adjacent Cluster Consolidation (ACC) mechanism that merges neighbor clusters in descriptor space. To further strengthen adaptation, we propose Relevance-guided Sample Retrieval (RSR), which selects the most target domain-relevant clusters for learning and integrates them into a Mean-Teacher self-supervised paradigm. Extensive experiments across CIFAR-10/100-C, ImageNet-C, and DomainNet demonstrate that MCM consistently outperforms prior methods under Practical TTA (PTTA) and achieves sustained robustness in recurring TTA. By delivering a memory structure that is more representative, scalable, and adaptive, MCM establishes multi-cluster memory as a practical and effective foundation for real-world test-time adaptation.

## 1 Introduction

Recent advances in machine learning, driven by vast training corpora and significant computational resources, have pushed model performance close to optimal within predefined application scenarios. However, in real-world deployments, models must contend with continuously evolving conditions that deviate from their training distributions. For example, a mature autonomous driving system must remain robust across diverse weather conditions and heterogeneous urban layouts (Yang et al., 2024; Yasarla et al., 2025). Similarly, in robotic manipulation, effective adaptation is crucial for enabling robotic arms to reliably identify and interact with objects of varying shapes and materials (Ren et al., 2023; Lu et al., 2024). Retraining for every unseen scenario is prohibitively costly in annotation and compute. This motivates test time adaptation (TTA) (Wang et al., 2021; Yuan et al., 2023; Hoang et al., 2024), which adapts models during deployment using only unlabeled test data, without revisiting the source training set. By enabling efficient in-situ updates under distribution shift, TTA offers a practical path to sustained robustness in ever-changing environments.

Early TTA methods updated models from the in-batch samples available at test time (Liang et al., 2020; Wang et al., 2021; Boudiaf et al., 2022). However, even at large batch sizes, these samples cover only a narrow and potentially biased slice of the target distribution. To address this limitation, subsequent methods introduced memory to accumulate high-confidence samples and approximate the target distribution more faithfully (Gong et al., 2022; Yuan et al., 2023; Kang et al., 2024). Nevertheless, most systems still manage memory with simple rules such as confidence or recency and treat it as an unstructured pool. We call this a *single-cluster memories*: samples are stored without organization that reflects multi-modal target structure. This leaves two central questions open: (1) do the retained samples provide a representative view of the target distribution, and (2) under continual shift, can the memory be updated quickly enough to preserve robustness?

To address the above questions, we first examine the essential properties of memory techniques for TTA, *i.e.*, *representativeness* and *adaptability*, by visualizing target distributions using Kernel Den-

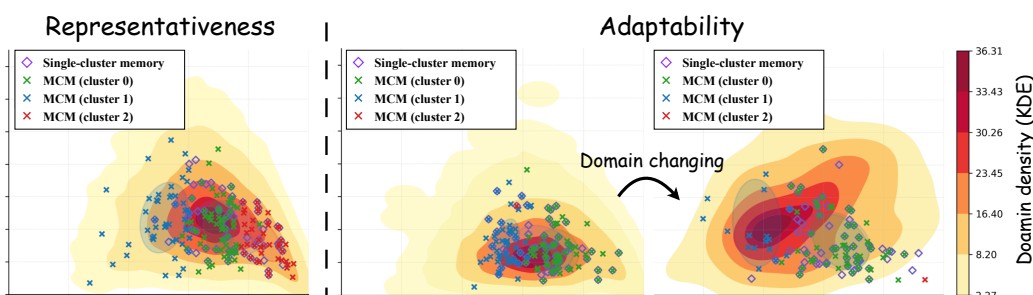

Figure 1: Visualization of two core properties of memory in TTA systems: *representativeness* (left) and *adaptability* (right). Diamond markers represent samples stored in the single-cluster memory (Yuan et al., 2023), which are concentrated in high-density regions. In contrast, colored cross markers represent samples stored in different clusters within our proposed Multi-Cluster Memory (MCM), demonstrating enhanced diversity and more effective adaptation to distribution shifts.

sity Estimation (KDE) (Parzen, 1962). KDE produces a smooth estimate of the data density from finite samples, where its level sets reveal high- and low-density regions and expose multiple modes that we colloquially refer to as *clusters*. In Figure 1, the left panel illustrates the representativeness issue: purple diamonds denote samples retained by previous work (Yuan et al., 2023) using single-cluster memory, which are concentrated in the central high-density region, failing to capture the broader distribution. The right panel highlights the restriction of adaptability under domain shifts, where the distribution evolves from left to right. Samples in a single-cluster memory remain confined to the high-density region of the previous domain, limiting their ability to adapt to the new distribution. While enlarging memory capacity can improve representativeness, single-cluster designs suffer from linearly growing management costs. Furthermore, under distribution shifts, these designs update samples sequentially, which restricts their ability to support rapid adaptation.

To address these limitations, we propose Multi-Cluster Memory (MCM), a novel memory management framework for test-time adaptation. The central idea of MCM is to structure memory into multiple clusters, where inter-cluster distances naturally expand the coverage of the sample distribution, yielding a more representative view of the target domain. Building on prior work (Huang & Belongie, 2017; Benz et al., 2021), we employ lightweight statistical descriptors—such as sample means and variances—as the organizing principle for clustering, enabling a memory representation that is both efficient and effective. When a test-time sample arrives, it is assigned to the cluster whose descriptor is closest; if it lies far from all existing clusters, a new cluster is created to preserve diversity. To prevent unbounded growth, we design Adjacent Cluster Consolidation (ACC), which merges the nearest clusters in descriptor space, ensuring bounded memory usage while maintaining distributional fidelity. Together, these components establish a memory architecture that is not only more representative, scalable, and robust than existing approaches, but also far more efficient in expanding memory capacity compared to the linearly growing cost within single-cluster designs.

On top of this architecture, we introduce Relevance-guided Sample Retrieval (RSR) to select reliable samples for adaptation. Specifically, the descriptors of the current mini-batch are compared against those of all clusters to identify the most domain-relevant clusters, which are then used for model updating. Following the Mean Teacher paradigm (Tarvainen & Valpola, 2017), adaptation proceeds in a self-supervised manner. By unifying principled memory management with relevance-guided retrieval, MCM selectively discards outdated clusters, rapidly adapts to new domains, and consistently retrieves high-quality samples for learning—ultimately achieving substantially better adaptability than conventional single-cluster memory designs.

We conduct extensive experiments across diverse image classification benchmarks under TTA settings, including CIFAR-10-C, CIFAR-100-C, ImageNet-C (Hendrycks & Dietterich, 2019), and DomainNet (Peng et al., 2019). Building on contemporary single-cluster memory–based TTA methods (Yuan et al., 2023; Hoang et al., 2024; Zhou et al., 2025), we replace their memory modules with our proposed MCM and consistently observe substantial performance gains. These results establish MCM as a scalable, plug-and-play component that can be seamlessly integrated into diverse TTA frameworks. We further evaluate the long-term robustness of MCM under the recurring

TTA setting (Hoang et al., 2024), where MCM continues to deliver stable improvements. Our key contributions are summarized as follows:

- We propose Multi-Cluster Memory, a scalable, plug-and-play memory management mechanism that strengthens both the *representativeness* and *adaptability* of TTA systems.
- By integrating statistical descriptors into the management process, we enable explicit and controllable organization of sample distributions within a multi-cluster memory.
- We demonstrate substantial and consistent performance improvements on diverse datasets under PTTA setting, achieving an average error reduction of 2.96% across 12 experimental configurations (ranging from 0.60% to 12.13%), and establish long-term robustness under recurring TTA setting with 2.5% improvement on CIFAR100-C.

## 2 RELATED WORK

**Evolution of Test-Time Adaptation Settings.** Test-time adaptation (TTA) emerged as a paradigm for adapting pre-trained models to target domains during inference without access to source data. Early methods (Mummadi et al., 2021; Wang et al., 2021) operated under a *fully TTA* setting, where the entire test set originates from a single fixed target domain. In this setup, all corruptions are treated uniformly, and adaptation proceeds directly from the source-trained model without accounting for temporal variation or domain evolution. Subsequently, CoTTA (Wang et al., 2022) extended this formulation to *continual TTA* (CTTA), where models must adapt to a sequence of evolving domains. To mitigate catastrophic forgetting, CoTTA introduces a stochastic restoration mechanism that intermittently resets the model to its source-pretrained state. Follow-up works (Zhu et al., 2024; Liu et al., 2024; Han et al., 2025) have further advanced this line of research.

To better approximate real-world conditions, samples from consecutive time steps exhibit inherent correlations, resulting in a non-i.i.d. sampling process. LAME (Boudiaf et al., 2022) was among the first to explicitly address this non-i.i.d. setting, while RoTTA (Yuan et al., 2023) further integrated it with CTTA, giving rise to the *Practical TTA* (PTTA) paradigm that more faithfully mirrors deployment scenarios. Building on this, recent advances such as PeTTA (Hoang et al., 2024) introduced the concept of *recurring TTA*, revealing that repeated adaptation cycles can eventually drive models toward collapse. In this work, we focus primarily on the PTTA setting, as it presents the most demanding challenge and offers the closest alignment with real-world deployment. Additionally, we evaluate under the recurring TTA setting to further assess long-term robustness.

**Memory-Based TTA Systems.** Memory has long been employed to preserve valuable information in artificial intelligence systems. Based on the nature of what is stored, memory can be broadly categorized into explicit memory (Rolnick et al., 2019; Song et al., 2023), implicit memory (Wu et al., 2022; Omidi et al., 2025; Tseng et al., 2025), and external information (Lewis et al., 2020; Wang et al., 2024). Explicit memory mechanisms have become indispensable for ensuring adaptation stability and mitigating catastrophic forgetting in practical TTA scenarios. Most existing TTA methods rely on single-cluster memory banks, which treat all stored samples as a homogeneous pool. For example, RoTTA (Yuan et al., 2023) employs heuristic scoring based on sample age and prediction uncertainty, while PeTTA (Hoang et al., 2024) extends this concept with persistent adaptation strategies. ECoTTA (Song et al., 2023) proposes self-distilled regularization to prevent model drift, and MemBN (Kang et al., 2024) emphasizes maintaining batch normalization statistics in memory. Additionally, ResiTTA (Zhou et al., 2025) introduces residual connections to enhance robustness in continual learning for TTA scenarios. Despite their contributions, these single-cluster approaches struggle to capture the complexity of manifold distributions, offering only an insensitive approximation of the target domains. This limitation underscores the need for more sophisticated memory mechanisms that can better represent and adapt to diverse and evolving target distributions.

## 3 METHODOLOGY

### 3.1 REVISITING MEMORY-BASED TEST-TIME ADAPTATION

Current memory-based TTA approaches (Yuan et al., 2023; Hoang et al., 2024) typically employ a single-cluster memory $\mathcal{M} = {x_i}_{i=1}^N$ that stores high-confidence samples. At each time step $t$,

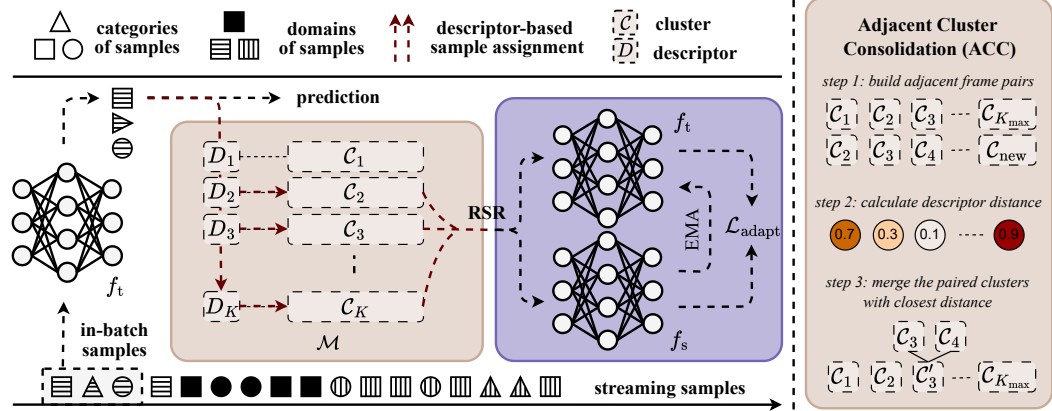

Figure 2: TTA system with Multi-Cluster Memory (MCM). Streaming samples are first predicted by the teacher model and assigned to clusters according to their descriptors. Relevance-guided Sample Retrieval (RSR) then supplies samples for adaptation. Through Adjacent Cluster Consolidation (ACC), MCM dynamically adjusts the clusters to prevent exceeding the predefined capacity.

the model first processes a mini-batch $\mathcal{B}_t$ to update the memory bank and subsequently adapts its parameters by minimizing

$$\mathcal{L}_{\text{adapt}} = \mathbb{E}_{x \sim \mathcal{M}} \left[ \mathcal{L}_{\text{cons}}(f_{\text{s}}(x), f_{\text{t}}(x)) \right], \tag{1}$$

where $f_{\text{s}}$ and $f_{\text{t}}$ denote the student and teacher networks, respectively. Crucially, adaptation relies exclusively on samples drawn from the memory bank $\mathcal{M}$ rather than the current batch $\mathcal{B}_t$. Following the Mean Teacher paradigm (Tarvainen & Valpola, 2017), the teacher produces pseudo-labels and is updated via an exponential moving average (EMA), which enhances training stability. While this single-cluster memory design alleviates the distributional narrowness inherent to pure in-batch adaptation, we emphasize that two fundamental limitations remain unresolved:

**Scalability Limits of Memory Capacity.** Although incorporating a memory mechanism broadens the coverage of the sample distribution in TTA systems, the extent of this coverage remains inherently limited. Moreover, contemporary single-cluster memory designs suffer from linearly increasing management costs, which in turn restrict their scalability.

**Sluggish Adaptation under Domain Shifts.** In continually evolving environments, the target distribution shifts from $P_{\mathcal{T}}^{(t)}$ to $P_{\mathcal{T}}^{(t+1)}$ across time steps. Owing to its homogeneous structure, a single-cluster memory struggles to preserve temporal diversity, as it cannot differentiate whether samples originate from past domains $P_{\mathcal{T}}^{(t-k)}$ or the current domain $P_{\mathcal{T}}^{(t)}$. This limitation induces a fundamental trade-off: aggressive replacement (small age weight) leads to catastrophic forgetting of prior knowledge, whereas conservative retention (large age weight) causes the memory to be dominated by outdated samples, diminishing its ability to represent the current distribution.

## 3.2 TEST-TIME ADAPTATION SYSTEM WITH MULTI-CLUSTER MEMORY

To improve the *representativeness and adaptability* of memory, we propose **Multi-Cluster Memory (MCM)** for TTA. As shown in Fig. 2, the memory bank is partitioned into up to $K_{\text{max}}$ clusters, $\mathcal{C}_1, \mathcal{C}_2, \ldots, \mathcal{C}_K$ with $K \leq K_{\text{max}}$, where each cluster $\mathcal{C}_k$ is formed by descriptor $D_k$ to capture distinct regions of the feature space. This design addresses two key limitations: (i) preserving diverse distributional patterns across clusters to avoid dominance by a single pattern, and (ii) enabling more efficient capacity expansion without the linearly growing management cost of single-cluster memory. With descriptor-based management, Adjacent Cluster Consolidation (ACC), and Relevance-guided Sample Retrieval (RSR), MCM supports efficient and effective test-time adaptation.

### 3.3 DESCRIPTOR-BASED MANAGEMENT

Our memory bank $\mathcal{M}$ is dynamically partitioned into $K$ clusters $\{\mathcal{C}_1, \mathcal{C}_2, ..., \mathcal{C}_K\}$ where $K \in \{1, 2, ..., K_{\max}\}$, starting from an empty state ($K = 0$). Each cluster $\mathcal{C}_k$ maintains up to $N$ samples, ensuring a maximum total capacity of $K_{\max} \times N$. To efficiently manage cluster assignment and consolidation, we characterize each sample $x$ by its channel-wise statistics descriptor:

$$d_x = [\mu_x^{(1)}, \sigma_x^{(1)}, ..., \mu_x^{(c)}, \sigma_x^{(c)}], \tag{2}$$

where $\mu_x^{(c)}$ and $\sigma_x^{(c)}$ denote the mean and variance of the $c$-th channel computed across spatial dimensions $H \times W$ of the feature map. Following previous work in test-time normalization (Tomar et al., 2024), these channel-wise statistics effectively capture domain shift characteristics while maintaining computational efficiency. Each cluster $\mathcal{C}_k$ is summarized by its centroid descriptor $D_k$, computed as the average of all member descriptors:

$$D_k = \frac{1}{|\mathcal{C}_k|} \sum_{x \in \mathcal{C}_k} d_x. \tag{3}$$

This lightweight descriptor enables efficient cluster operations in online adaptation scenarios.

**Sample Assignment.** Upon arrival of a new sample $x_t$ at time $t$, we perform cluster assignment followed by selective insertion or replacement to maintain high intra-cluster density. We compute the Euclidean distance between the sample descriptor $d_{x_t}$ and all existing cluster centroids:

$$k^* = \underset{k \in \{1, ..., K\}}{\arg\min} \|d_{x_t} - D_k\|_2. \tag{4}$$

If the minimum distance exceeds the threshold $\tau$, indicating that $x_t$ lies far from all existing clusters, a new cluster is spawned as $\mathcal{C}_{K+1} = \{x_t\}$. Otherwise, $x_t$ is assigned to the nearest cluster $\mathcal{C}_{k^*}$.

**Sample Replacement.** When the target cluster $\mathcal{C}_{k^*}$ reaches capacity ($|\mathcal{C}_{k^*}| = N$), we employ a heuristic scoring function to identify the least valuable sample for replacement. Building upon the scoring function from Yuan et al. (2023), we incorporate descriptor distance as an additional term:

$$H(x) = \lambda_t \cdot \frac{1}{1 + \exp(-A_x/N)} + \lambda_u \cdot \frac{U_x}{\log N_C} + \lambda_d \cdot \|d_x - D_{k^*}\|_2, \tag{5}$$

where $A_x$ denotes the age of sample $x$ (i.e., the number of steps since insertion), $U_x$ represents its prediction entropy, $N_C$ is the number of classes, and the third term quantifies the distance to the cluster centroid. The sample with the highest score is then replaced by $x_t$. By extending RoTTA's timeliness and uncertainty criteria with a spatial distance term, this strategy ensures that clusters preserve not only temporal relevance and prediction confidence but also spatial compactness.

### 3.4 ADJACENT CLUSTER CONSOLIDATION

When the number of clusters reaches $K_{\max}$, a consolidation step is triggered to merge the most similar clusters. To balance efficiency with the sequential growth of clusters, we adopt the consolidation strategy of Song et al. (2024), restricting candidates to adjacent cluster pairs in the creation sequence, as these are more likely to correspond to related distributions. For each adjacent pair $(\mathcal{C}i, \mathcal{C}i+1)$, we compute their centroid distance $\Delta_{i,i+1} = \|D_i - D_{i+1}\|_2$ and merge the pair with the minimum distance. The consolidation process unifies all samples from both clusters into a single pool and retains the $N$ samples with the lowest prediction uncertainty. Formally, the merged cluster is defined as

$$\mathcal{C}_{\text{merged}} = \begin{cases} \mathcal{C}_i \cup \mathcal{C}_j & \text{if } |\mathcal{C}_i \cup \mathcal{C}_j| \leq N \\ \text{top-}K(\{x \in \mathcal{C}_i \cup \mathcal{C}_j : U_x \text{ ascending}\}, N) & \text{otherwise} \end{cases}. \tag{6}$$

After cluster merging, we reconstruct the class-wise structure following Yuan et al. (2023) to maintain balanced representation across categories. The merged cluster descriptor is then updated. This strategy enhances memory efficiency while preserving high-confidence samples and class diversity.

Table 1: Overall Practical Test-time Adaptation (PTTA) error rates (%) on CIFAR10-C, CIFAR100-C, ImageNet-C (Hendrycks & Dietterich, 2019), and DomainNet (Peng et al., 2019) (severity 5). Lower is better. Numbers in parentheses indicate improvement margins over their respective baseline. † denotes results from our implementation as the original paper did not report on this dataset.

| Method | Venue | CIFAR10-C | CIFAR100-C | ImageNet-C | DomainNet |
|---|---|---|---|---|---|
| Source | – | 43.50 | 46.40 | 82.00 | – |
| BN | CoRR'20 | 75.20 | 52.90 | – | – |
| PL | ICML'13 | 82.90 | 88.90 | – | – |
| TENT | ICLR'21 | 86.00 | 92.80 | – | – |
| LAME | CVPR'22 | 39.50 | 40.50 | 80.90 | – |
| CoTTA | CVPR'22 | 83.20 | 52.20 | 98.60 | – |
| NOTE | NeurIPS'22 | 31.10 | 73.80 | – | – |
| RDumb | NeurIPS'23 | 31.10 | 36.70 | 72.20 | 44.30 |
| ROID | WACV'24 | 72.70 | 76.40 | 62.70 | – |
| TRIBE | AAAI'24 | **15.30** | 33.80 | 63.60 | – |
| RoTTA | CVPR'23 | 25.20 | 35.00 | 68.30 | 44.30 |
| + MCM | – | 22.59 (-2.61) | 33.75 (-1.25) | 67.46 (-0.84) | 42.53 (-1.77) |
| PeTTA | NeurIPS'24 | 24.30 | 35.80 | 65.30 | 43.80 |
| + MCM | – | 21.55 (-2.75) | 33.04 (-2.76) | **60.30** (-5.00) | 42.80 (-1.00) |
| ResiTTA | ICASSP'25 | 22.80 | 32.50 | 69.40 | 54.76† |
| + MCM | – | 20.69 (-2.11) | **31.90** (-0.60) | 66.65 (-2.75) | **42.63** (-12.13) |

### 3.5 Relevance-Guided Sample Retrieval

During adaptation, we selectively retrieve samples from clusters most relevant to the current batch $\mathcal{B}t$. Each cluster's relevance score is computed as the average descriptor distance to all in-batch samples. The $N_S$ clusters with the lowest scores (i.e., highest similarity) form the retrieval set $\mathcal{M}$retrieve, from which samples are drawn for model updates. Following established TTA practices (Yuan et al., 2023; Hoang et al., 2024), we employ the Mean Teacher framework (Tarvainen & Valpola, 2017) with consistency regularization as defined in equation 1. The novelty of our approach lies not in the adaptation mechanism itself, but in the construction of $\mathcal{M}_{\text{retrieve}}$ through our MCM, which yields memory samples that are both more representative of the target distribution and more adaptable to continual domain shifts than single-cluster designs.

**Discussion.** With respect to adaptability in evolving domains, single-cluster memory suffers from sequential sample updates, resulting in an inefficient transfer process. In contrast, our proposed MCM, empowered by ACC and RSR management strategies, enables rapid cluster deletion and creation, followed by targeted sample retrieval from the most relevant clusters. Collectively, these capabilities enhance the representativeness and adaptability of memory under continual domain shifts.

## 4 Experiments

### 4.1 Setup and Protocols

**Datasets and Metrics.** We evaluate our method on four benchmark datasets under the Practical Test-Time Adaptation (PTTA) setting (Yuan et al., 2023). For CIFAR10-C, CIFAR100-C, and ImageNet-C (Hendrycks & Dietterich, 2019), we adopt severity level 5. We further evaluate on DomainNet (Peng et al., 2019), using 126 categories across four domains for the transfer task *real* $\rightarrow$ clipart, painting, sketch. The performance is measured by the mean classification error.

**Implementation Details.** Following Hoang et al. (2024), we use WideResNet-28 (Zagoruyko & Komodakis, 2016) for CIFAR10-C, ResNeXt-29 (Xie et al., 2017) for CIFAR100-C, and ResNet-50 (Croce et al., 2021) for ImageNet-C and DomainNet. All experiments are conducted on a single NVIDIA RTX 4090 GPU. Following Yuan et al. (2023), we adopt the PTTA protocol and use Dirichlet distribution to simulate correlative sampling with parameter $\delta = 0.1$ for CIFAR10-C and CIFAR100-C to simulate temporal correlation. In MCM, we set $K_{\max} = 1$ for CIFAR10-C and

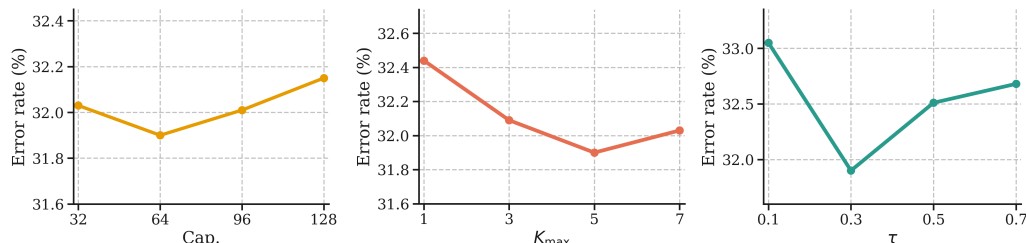

Figure 3: These figures demonstrate the effectiveness of the hyper-parameters, *i.e.*, Cap., $K_{\max}$, and $\tau$. We conduct the analysis on CIFAR100-C under PTTA setting and integrate ResiTTA with MCM.

$K_{\max} = 5$ for more complex datasets (CIFAR100-C, ImageNet-C, and DomainNet). A detailed analysis of $K_{\max}$ is provided in the Section 4.3 content and further expanded in the Appendix C.

**Baselines.** We compare with: Source (no adaptation), BN (Nado et al., 2020), PL (Lee et al., 2013), TENT (Wang et al., 2021), LAME (Boudiaf et al., 2022), CoTTA (Wang et al., 2022), NOTE (Gong et al., 2022), RDumb (Press et al., 2023), ROID (Marsden et al., 2024), and TRIBE (Su et al., 2024). For baselines with publicly available implementations, we adopt their original parameter settings. Hyperparameter choices are kept as close as possible to the original selections.

## 4.2 MAIN RESULTS

Table 1 reports the performance comparison under the PTTA setting across CIFAR10-C, CIFAR100-C, ImageNet-C, and DomainNet. To demonstrate both effectiveness and generalizability, we integrate MCM with contemporary memory-based TTA methods—RoTTA, PeTTA, and ResiTTA.

**Consistent Gains Across Baselines.** Our approach yields consistent gains across all baselines methods across datasets. When combined with PeTTA, it achieves the best performance on ImageNet-C (60.30%), marking a substantial improvement of 5.00%. Likewise, with ResiTTA, it delivers the top results on CIFAR100-C (31.90%) and DomainNet (42.63%). The results demonstrate that MCM effectively overcomes the fundamental limitations of single-cluster memory approaches.

**Advantage on Complex Distributions.** The performance gains are most pronounced on challenging benchmarks. On ImageNet-C and DomainNet, which feature larger label spaces and greater distributional diversity, our method achieves average improvements of 2.86% and 4.97% respectively across the three baselines. These results highlight that the proposed MCM is particularly advantageous in complex domains, where a single-cluster design fails to capture the full view of the target distribution. Nevertheless, while our approach consistently improves over baselines, MCM can only yield competitive performance on relatively less complex datasets such as CIFAR10-C.

## 4.3 ANALYSIS OF HYPERPARAMETER SENSITIVITY

As shown in Figure 3, we conduct an ablation study on the key hyperparameters of our proposed MCM, namely the capacity of a single cluster ($Cap.$), the maximum number of clusters ($K_{\max}$), and the distance threshold ($\tau$) for cluster creation. The analysis is performed on CIFAR100-C using ResiTTA equipped with MCM. Extended results are provided in the Appendix C.

**Effectiveness of Cluster Capacity.** In this analysis, we fix $K_{\max} = 5$ and $\tau = 0.3$. As shown in the left panel of Figure 3, the error rate first rises as the cluster capacity increases, but after exceeding 64, it reverses and declines with further expansion. This indicates that scaling capacity alone is ineffective; only when combined with proper inter-cluster management strategies can the overall effectiveness of TTA be substantially improved.

**Impact of the Number of the Clusters.** In this analysis, we fix Cap. = 64 and $\tau = 0.3$. As shown in the middle panel of Figure 3, the optimal choice for the maximum number of clusters is $K_{\max} = 5$. Consistent with the earlier findings, the trending of error rate indicates that Cap. and $K_{\max}$ must be jointly considered, as their interplay is crucial for achieving effective adaptation.

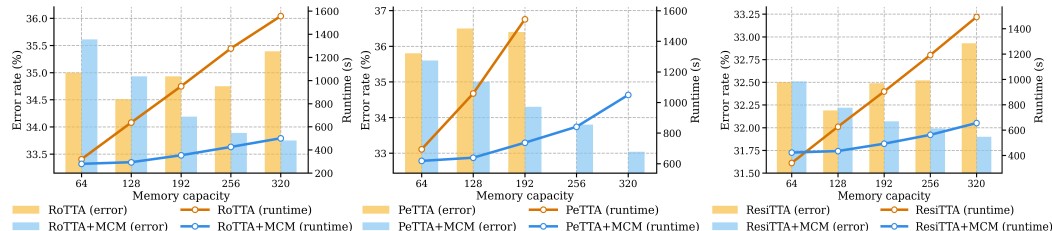

Figure 4: Runtime–error comparisons for RoTTA, PeTTA, and ResiTTA with and without MCM. The x-axis denotes the total number of samples stored in memory. In MCM, the per-cluster size is fixed at 64, and different totals are obtained by varying the number of clusters. Bars show the *error rate* (left y-axis), and lines show the *runtime* (right y-axis, seconds) for whole CIFAR100-C dataset. As shown in middle panel, PeTTA with 256 and 320 samples encountered out-of-memory errors.

**Effect of Distance Threshold.** In this analysis, we fix Cap. $= 64$ and $K_{max} = 5$. As shown in the right panel of Figure 3, the best error rate is achieved with threshold of $\tau = 0.3$. Notably, empirical evidence from single-cluster memory-based TTA methods (Yuan et al., 2023; Hoang et al., 2024) suggests that thresholds above $0.5$ generally yield stronger performance. We interpret this discrepancy as evidence that the multi-cluster design decomposes the complex target distribution into finer-grained regions, each managed by a dedicated cluster. Under this design, applying a stricter threshold $\tau$ enhances cluster-level representativeness and ensures more reliable adaptation.

## 4.4 SCALING MEMORY CAPACITY WITH EFFICIENCY

To ensure that the gains of MCM are not attributed merely to enlarging memory capacity, we conducted a controlled experiment by integrating MCM with RoTTA, PeTTA, and ResiTTA on CIFAR100-C. In this setting, conventional single-cluster memories were scaled by directly increasing the number of stored samples, whereas MCM increased its maximum number of clusters. Both strategies were compared under the same total number of stored samples.

As shown in Figure 4, simply enlarging memory in existing methods yields marginal accuracy improvements but incurs substantial computational overhead. For instance, when RoTTA's memory bank is scaled from 64 to 320 samples, the error rate remains almost unchanged (35.0% to 35.4%), while runtime escalates dramatically from 280s to 1556s due to the linear growth of management costs. This trend is consistent across all baselines, suggesting that naively storing more samples does not capture the distributional diversity necessary for stronger adaptation.

In contrast, our proposed MCM organization achieves a more favorable trade-off between accuracy and efficiency. By structuring 320 samples into 5 clusters of 64, RoTTA+MCM reduces the error rate to 33.8% while requiring only 501s. The structured design enables memory to be utilized more effectively, amplifying representativeness without incurring linear cost growth.

Direct comparisons under equal capacities further highlight MCM's advantage: with 320 samples, RoTTA+MCM achieves 33.8% error in 501s, whereas standard RoTTA yields 35.4% error in 1556s. These results demonstrate that MCM's improvements derive from its principled memory structure rather than raw capacity, underscoring its practical value for real-world deployment where both accuracy and efficiency are critical.

## 4.5 LONG-TERM ROBUSTNESS

In addition to the PTTA evaluation, we further assess our approach under the more challenging *recurring TTA* protocol introduced by Hoang et al. (2024). In this setting, the model repeatedly encounters the same sequence of corruptions across multiple adaptation rounds, providing a stringent test of long-term stability and resilience against catastrophic forgetting. We integrate our proposed MCM into PeTTA and conduct experiments on CIFAR100-C within this recurring TTA regime. As shown in Table 2, MCM achieves consistent and substantial performance gains, with improvements that accumulate as the number of adaptation rounds increases. These findings demonstrate that, when built upon PeTTA, MCM not only preserves baseline robustness but also yields steady

Table 2: **CIFAR100 → CIFAR100-C, Recurring TTA (severity 5).** Columns 1–20 list the classification error rate for each successive revisit to the corruption stream; *Avg* is the mean over all 20 visits. Results are obtained with a ResNeXt-29 backbone and the official RobustBench preprocessing. **Bold** denotes the best method and underlined the second best in every column.

| Method | 1 | 2 | 3 | 4 | 5 | 6 | 7 | 8 | 9 | 10 | 11 | 12 | 13 | 14 | 15 | 16 | 17 | 18 | 19 | 20 | Avg |
|---|---|---|---|---|---|---|---|---|---|---|---|---|---|---|---|---|---|---|---|---|---|
| Source | | | | | | | | | | | 46.5 | | | | | | | | | | |
| LAME | | | | | | | | | | | 40.5 | | | | | | | | | | |
| CoTTA | 53.4 | 58.4 | 63.4 | 67.6 | 71.4 | 74.9 | 78.2 | 81.1 | 84.0 | 86.7 | 88.8 | 90.7 | 92.3 | 93.5 | 94.7 | 95.6 | 96.3 | 97.0 | 97.3 | 97.6 | 83.1 |
| EATA | 88.5 | 95.0 | 96.8 | 97.3 | 97.4 | 97.2 | 97.2 | 97.3 | 97.4 | 97.5 | 97.5 | 97.5 | 97.6 | 97.7 | 97.7 | 97.7 | 97.8 | 97.8 | 97.7 | 97.7 | 96.9 |
| RMT | 50.5 | 48.6 | 47.9 | 47.4 | 47.3 | 47.1 | 46.9 | 46.9 | 46.6 | 46.8 | 46.7 | 46.5 | 46.5 | 46.6 | 46.5 | 46.5 | 46.5 | 46.5 | 46.5 | 46.5 | 47.1 |
| MECTA | 44.8 | 44.3 | 44.6 | 43.1 | 44.8 | 44.2 | 44.4 | 43.8 | 43.8 | 43.9 | 44.6 | 43.8 | 44.4 | 44.6 | 43.9 | 44.2 | 43.8 | 44.4 | 44.9 | 44.2 | 44.2 |
| RoTTA | 35.5 | 35.2 | 38.5 | 41.9 | 45.3 | 49.2 | 52.0 | 55.2 | 58.1 | 61.5 | 64.6 | 67.5 | 70.7 | 73.2 | 75.4 | 77.1 | 79.2 | 81.5 | 82.8 | 84.5 | 61.4 |
| RDumb | 36.7 | 36.7 | 36.6 | 36.6 | 36.7 | 36.8 | 36.7 | 36.5 | 36.6 | 36.5 | 36.7 | 36.6 | 36.5 | 36.7 | 36.5 | 36.6 | 36.6 | 36.7 | 36.6 | 36.5 | 36.6 |
| ROID | 76.4 | 76.4 | 76.2 | 76.2 | 76.3 | 76.1 | 75.9 | 76.1 | 76.3 | 76.3 | 76.6 | 76.3 | 76.8 | 76.7 | 76.6 | 76.3 | 76.2 | 76.0 | 75.9 | 76.0 | 76.3 |
| TRIBE | 33.8 | 33.3 | 35.3 | 34.9 | 35.3 | 35.1 | 37.1 | 37.2 | 37.2 | 39.1 | 39.2 | 41.1 | 41.0 | 43.1 | 45.1 | 45.1 | 45.0 | 44.9 | 44.9 | 44.9 | 39.6 |
| PeTTA | 35.8 | 34.4 | 34.7 | 35.0 | 35.1 | 35.1 | 35.2 | 35.3 | 35.3 | 35.3 | 35.2 | 35.3 | 35.2 | 35.2 | 35.1 | 35.2 | 35.2 | 35.2 | 35.2 | 35.2 | 35.1 |
| PeTTA + MCM | **33.8** | **33.8** | **33.0** | **33.0** | **33.1** | **33.9** | **33.9** | **33.9** | **32.7** | **32.7** | **32.7** | **32.8** | **32.7** | **32.7** | **32.7** | **32.6** | **32.6** | **32.6** | **32.6** | **32.5** | **32.6** |

long-term benefits through successive adaptations. We attribute these gains to the enhanced representativeness of MCM's memory, which supplies higher-quality learning samples throughout the adaptation process. Additional results under the recurring TTA setting are provided in Appendix B.

## 5 DISCUSSION

The MCM architecture reveals fundamental insights into how sample organization affects test-time adaptation. Our analysis across multiple benchmarks demonstrates that the performance gains arise not from increased storage capacity, but from principled structural changes to memory management.

**Limitations.** While effective, MCM has several limitations: (i) it introduces additional computational overhead from descriptor computation and cluster management; (ii) it exhibits sensitivity to a small set of hyperparameters (e.g., $\tau$ and the consolidation rule), which may require light tuning across datasets; and (iii) the usage of the descriptors (*i.e.*, statistic of samples) embodies an inductive bias toward shifts that primarily manifest in first- or second-order feature statistics. Moreover, our evaluations are confined to image classification; extending MCM to other modalities and to more severe distribution shifts remains an important direction for future work.

**Future Work.** We regard memory as an indispensable component of the test-time adaptation process and foresee several promising directions for its advancement: (i) Existing memory mechanisms, including MCM, typically store raw samples as the primary content, which imposes substantial storage costs. This challenge becomes even more pronounced when extending TTA to large language models or multi-modal architectures. A natural next step is to employ compact feature representations as the storage unit, thereby improving both efficiency and scalability; (ii) In its current form, MCM limits consolidation to adjacent clusters due to sequential updates, implicitly treating clusters as a linear list. However, clusters are inherently unordered and related only through descriptor-based distances. A graph-based management strategy thus represents a compelling future direction, offering both greater efficiency and a closer alignment with the underlying structure.

## 6 CONCLUSION

We introduced Multi-Cluster Memory (MCM), a structured memory framework for test-time adaptation that leverages descriptor-guided clustering, Adjacent Cluster Consolidation, and Relevance-guided Sample Retrieval to enhance representativeness and adaptability. Extensive experiments across CIFAR-10-C, CIFAR-100-C, ImageNet-C, and DomainNet under the PTTA protocol confirm consistent improvements over state-of-the-art baselines, while recurring TTA evaluations further demonstrate long-term stability. Our analysis reveals that performance gains arise not from enlarging memory capacity, but from principled organization and retrieval, highlighting memory as a core architectural element of adaptation. Although MCM introduces modest overhead and limited consolidation scope, these trade-offs are outweighed by robustness and scalability. Looking forward, adopting compact feature-level storage and graph-based cluster management promises to further improve efficiency and alignment with the underlying data structure.

## ETHICS STATEMENT

This work adheres to the ICLR Code of Ethics.[1] Our study focuses on developing a novel memory mechanism for test-time adaptation (TTA), termed Multi-Cluster Memory (MCM). The research does not involve human subjects, personally identifiable information, or sensitive data, and all datasets used (CIFAR10-C, CIFAR100-C, ImageNet-C, DomainNet) are publicly available benchmarks with established community usage. We acknowledge that improvements in TTA methods may indirectly affect safety-critical applications such as autonomous driving and decision-making systems. To mitigate risks, we ensure reproducibility and transparency by providing clear methodological descriptions and adhering to community practices. No part of this work is intended to enable harmful or malicious applications, and we emphasize that responsible deployment in real-world scenarios requires careful evaluation of safety, fairness, and robustness.

## REPRODUCIBILITY STATEMENT

We have made extensive efforts to ensure the reproducibility. All implementation details of the proposed Multi-Cluster Memory (MCM) framework, including the memory partitioning algorithm, cluster management strategies (ACC and RSR), and training protocols, are fully described in Section 3. The experimental setup, including datasets (CIFAR10-C, CIFAR100-C, ImageNet-C, and DomainNet), evaluation metrics, and PTTA protocol, is detailed in Section 4. Ablation studies on key hyper-parameters (cluster capacity, maximum number of clusters $K_{max}$, and distance threshold $\tau$) are provided in Section 4.3 and Appendix C to validate robustness. Moreover, we will release anonymized source code and configuration files as supplementary material to enable independent verification of our results. These resources collectively ensure that our reported results can be faithfully reproduced and extended by the community.

---

[1]https://iclr.cc/public/CodeOfEthics

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

# From Single Bank to Multi-Clusters:
# Memory Architectures for Test-Time Adaptation
# Technical Appendices

## A   LLM USAGE STATEMENT

In preparing this work, we employed a large language model (LLM) solely as an auxiliary tool for improving writing clarity and logical organization. The LLM was used to refine the presentation of ideas, polish the language, and ensure consistency in terminology across the paper. Importantly, the LLM was not involved in research ideation, experimental design, algorithm development, or data analysis. All scientific contributions, technical innovations, and experimental results are entirely the work of the authors. The authors take full responsibility for the content of this paper.

## B   RECURRING TEST-TIME ADAPTATION

In addition to the standard PTTA evaluation, we assess our method's robustness under the more challenging recurring TTA setting introduced by PeTTA (Hoang et al., 2024). In this protocol, the model repeatedly encounters the same sequence of corruptions across multiple rounds, providing insights into long-term adaptation stability and resistance to catastrophic forgetting.

Table 3 presents results on CIFAR10-C over 20 consecutive rounds. While most baseline methods exhibit severe performance degradation—with CoTTA, EATA, and MECTA reaching error rates above 85% by round 20—our multi-cluster memory architecture maintains stable performance. When integrated with PeTTA, our approach achieves an average error of 19.5% across all rounds, representing a 3.3% improvement over PeTTA alone. Notably, our method demonstrates remarkable consistency, with error rates fluctuating within a narrow 1.6% range (18.7%–20.4%) throughout the 20 rounds, compared to PeTTA's baseline variance of 0.7% (22.3%–23.0%). This stability suggests that the multi-cluster organization effectively balances the retention of historical knowledge with adaptation to recurring patterns.

The performance gap between our method and baselines widens progressively across rounds. While we observe only a 1.9% improvement in round 1, this advantage grows to 3.0% by round 20, indicating that our approach becomes increasingly beneficial under prolonged adaptation scenarios. This trend validates our hypothesis that organizing memory into multiple clusters prevents the homogenization that plagues single-cluster approaches during extended adaptation periods.

Table 3: **CIFAR-10 → CIFAR-10-C, Recurring TTA (severity 5).** Columns 1–20 report the classification error rates (%, lower is better) at each revisit to the corruption stream, with *Avg* denoting the mean over all 20 visits. All methods use a WideResNet-28 backbone obtained from ROBUST-BENCH, along with its official preprocessing pipeline.

| Method | 1 | 2 | 3 | 4 | 5 | 6 | 7 | 8 | 9 | 10 | 11 | 12 | 13 | 14 | 15 | 16 | 17 | 18 | 19 | 20 | Avg |
|---|---|---|---|---|---|---|---|---|---|---|---|---|---|---|---|---|---|---|---|---|---|
| Source | | | | | | | | | | | 43.5 | | | | | | | | | | |
| LAME | | | | | | | | | | | 31.1 | | | | | | | | | | |
| CoTTA | 82.2 | 85.6 | 87.2 | 87.8 | 88.2 | 88.5 | 88.7 | 88.7 | 88.9 | 88.9 | 88.9 | 89.2 | 89.2 | 89.2 | 89.1 | 89.2 | 89.2 | 89.1 | 89.3 | 89.3 | 88.3 |
| EATA | 81.6 | 87.0 | 88.7 | 88.7 | 88.9 | 88.7 | 88.6 | 89.0 | 89.3 | 89.6 | 89.5 | 89.6 | 89.7 | 89.7 | 89.3 | 89.6 | 89.6 | 89.8 | 89.9 | 89.4 | 88.8 |
| RMT | 77.5 | 76.9 | 76.5 | 75.8 | 75.5 | 75.5 | 75.4 | 75.4 | 75.5 | 75.3 | 75.5 | 75.6 | 75.5 | 75.5 | 75.7 | 75.6 | 75.7 | 75.6 | 75.7 | 75.8 | 75.8 |
| MECTA | 72.2 | 82.0 | 85.2 | 86.3 | 87.0 | 87.3 | 87.3 | 87.5 | 88.1 | 88.8 | 88.9 | 88.9 | 88.6 | 89.1 | 88.7 | 88.8 | 88.5 | 88.6 | 88.3 | 88.8 | 86.9 |
| RoTTA | 24.6 | 25.5 | 29.6 | 33.6 | 38.2 | 42.8 | 46.2 | 50.6 | 52.2 | 54.1 | 56.5 | 57.5 | 59.4 | 60.2 | 61.7 | 63.0 | 64.8 | 66.1 | 68.2 | 70.3 | 51.3 |
| RDumb | 31.1 | 32.1 | 32.3 | 31.6 | 31.9 | 31.8 | 31.8 | 31.9 | 31.9 | 32.1 | 31.7 | 32.0 | 32.5 | 32.0 | 31.9 | 31.6 | 31.9 | 31.4 | 32.3 | 32.4 | 31.9 |
| ROID | 72.7 | 72.6 | 73.1 | 72.4 | 72.7 | 72.8 | 72.7 | 72.7 | 72.9 | 72.8 | 72.9 | 72.9 | 72.8 | 72.5 | 73.0 | 72.8 | 72.5 | 72.5 | 72.7 | 72.7 | 72.7 |
| TRIBE | **15.3** | **16.6** | **16.6** | **16.3** | **16.7** | **17.0** | **17.3** | **17.4** | **17.4** | **18.0** | **17.9** | **18.0** | **17.9** | **18.6** | **18.2** | **18.8** | **18.0** | **18.2** | **18.4** | **18.0** | **17.5** |
| PeTTA | 24.3 | 23.0 | 22.6 | 22.4 | 22.4 | 22.5 | 22.3 | 22.5 | 22.8 | 22.8 | 22.6 | 22.7 | 22.7 | 22.9 | 22.6 | 22.7 | 22.6 | 22.8 | 22.9 | 23.0 | 22.8 |
| PeTTA+MCM | 21.7 | 21.0 | 20.4 | 19.8 | 20.7 | 20.1 | 20.9 | 20.2 | 20.1 | 20.4 | 20.5 | 20.3 | 20.1 | 20.0 | 19.8 | 20.1 | 20.0 | 20.3 | 20.5 | 20.7 | 20.4 |

## C   IMPACT OF MEMORY ARCHITECTURE PARAMETERS

To further understand the design choices in our multi-cluster memory architecture, we conduct detailed ablation studies using our best-performing configuration (PeTTA+Ours) on CIFAR100-C. Table 4 presents comprehensive experiments examining two critical hyperparameters: the maximum

Table 4: **Ablation study for design in multi-cluster memory bank on CIFAR100 $\rightarrow$ CIFAR100-C (severity 5).** We investigate the impact of the per-cluster capacity $N$ and the value of the base threshold $\tau$ used during sample assignment on the performance of test-time adaptation.

| $K_{max}$ | $N$ | 1 | 2 | 3 | 4 | 5 | 6 | 7 | 8 | 9 | 10 | 11 | 12 | 13 | 14 | 15 | 16 | 17 | 18 | 19 | 20 | Avg |
|---|---|---|---|---|---|---|---|---|---|---|---|---|---|---|---|---|---|---|---|---|---|---|
| 1 | 128 | 34.5 | 33.8 | 34.0 | 34.0 | 34.1 | 34.1 | 34.1 | 34.0 | 34.0 | 34.1 | 34.0 | 34.0 | 34.0 | 33.9 | 34.0 | 33.9 | 33.9 | 34.0 | 34.0 | 33.8 | 34.0 |
| 5 | 16 | 33.4 | 33.3 | 33.6 | 33.6 | 33.6 | 33.3 | 33.4 | 33.4 | 33.3 | 33.4 | 33.5 | 33.3 | 33.3 | 33.2 | 33.2 | 33.3 | 33.3 | 33.2 | 33.1 | 33.1 | 33.4 |
| 5 | 32 | 33.3 | 33.1 | 33.3 | 33.3 | 33.4 | 33.2 | 33.2 | 33.2 | 33.0 | 33.0 | 33.0 | 33.1 | 33.0 | 33.0 | 33.0 | 32.9 | 32.9 | 32.9 | 32.9 | 32.8 | 33.1 |
| 5 | 64 | 33.0 | 32.9 | 33.0 | 33.2 | 33.3 | 33.1 | 33.1 | 33.0 | 32.9 | 33.0 | 33.0 | 33.0 | 32.9 | 32.8 | 32.8 | 33.0 | 32.8 | 32.8 | 32.8 | 32.7 | 32.9 |
| 5 | 128 | 33.1 | 32.9 | 33.0 | 33.2 | 33.2 | 33.0 | 33.3 | 33.0 | 33.0 | 33.1 | 32.9 | 33.0 | 33.0 | 32.9 | 32.9 | 33.0 | 32.9 | 32.9 | 32.8 | 32.9 | 33.0 |

| $\tau$ | | 1 | 2 | 3 | 4 | 5 | 6 | 7 | 8 | 9 | 10 | 11 | 12 | 13 | 14 | 15 | 16 | 17 | 18 | 19 | 20 | Avg |
|---|---|---|---|---|---|---|---|---|---|---|---|---|---|---|---|---|---|---|---|---|---|---|---|
| 0.1 | | 34.8 | 34.4 | 34.5 | 34.6 | 34.7 | 34.6 | 34.8 | 34.7 | 34.7 | 34.5 | 34.6 | 34.4 | 34.3 | 34.4 | 34.2 | 34.3 | 34.4 | 34.3 | 34.1 | 34.1 | 34.5 |
| 0.3 | | 33.3 | 33.1 | 33.3 | 33.3 | 33.4 | 33.2 | 33.2 | 33.2 | 33.0 | 33.0 | 33.0 | 33.1 | 33.0 | 33.0 | 33.0 | 32.9 | 33.0 | 32.9 | 32.9 | 32.8 | 33.1 |
| 0.5 | | 35.1 | 34.4 | 34.4 | 34.5 | 34.4 | 34.4 | 34.4 | 34.4 | 34.1 | 34.2 | 34.1 | 34.2 | 34.0 | 34.1 | 34.2 | 34.3 | 34.3 | 34.1 | 34.0 | 34.2 | 34.3 |
| 0.7 | | 35.5 | 34.6 | 34.9 | 35.0 | 35.0 | 34.9 | 34.9 | 34.8 | 34.9 | 34.9 | 34.8 | 34.9 | 34.8 | 34.7 | 34.9 | 35.0 | 34.8 | 34.8 | 34.8 | 34.7 | 34.9 |

number of clusters ($K_{max}$) and the distance threshold ($\tau$) for cluster creation. All experiments in this section build upon the PeTTA+Ours framework, isolating the impact of memory architecture parameters while maintaining other components constant.

For cluster capacity analysis, we first compare single-cluster ($K_{max} = 1$) versus multi-cluster ($K_{max} = 5$) configurations with fixed capacity $N = 128$ under the PeTTA+Ours framework. The multi-cluster design achieves 33.0% error compared to 34.0% for single-cluster, demonstrating a consistent 1.0% improvement even within our already enhanced system. This validates that the benefits of multi-cluster organization are complementary to PeTTA's persistent adaptation strategy. Within the multi-cluster configuration, we observe that performance improves monotonically with increased per-cluster capacity: from 33.4% error at $N = 16$ to 32.9% at $N = 64$. However, further increasing to $N = 128$ yields marginal returns (33.0%), suggesting that moderate cluster sizes (32-64 samples) achieve an optimal balance between diversity and computational efficiency in the PeTTA+Ours framework.

The distance threshold $\tau$ critically influences cluster formation dynamics within our integrated system. Our experiments with PeTTA+Ours reveal that $\tau = 0.3$ achieves optimal performance (33.1% error), significantly outperforming both conservative ($\tau = 0.1$, 34.5% error) and aggressive ($\tau = 0.7$, 34.9% error) thresholds. A small threshold creates excessive fragmentation by spawning clusters for minor distributional variations, while a large threshold fails to capture meaningful diversity by forcing dissimilar samples into the same cluster. The optimal value of 0.3 in the PeTTA+Ours configuration suggests that successful adaptation requires distinguishing between genuine distributional modes while avoiding over-segmentation of continuous distributions, particularly when combined with PeTTA's persistent adaptation mechanisms.

## D    CORRUPTION-SPECIFIC PERFORMANCE ANALYSIS

Tables 5 and 6 present detailed corruption-wise performance under the PTTA protocol, revealing interesting patterns in how our multi-cluster memory handles different types of distribution shifts.

On CIFAR10-C, our method achieves particularly strong improvements on corruptions that induce geometric transformations (elastic: 18.6% vs 24.6% for ResiTTA) and weather-related effects (frost: 15.7% vs 18.5%). These corruptions often create distinct visual patterns that benefit from separate cluster representations. Conversely, the improvement is minimal for noise-based corruptions (impulse, gaussian), where the corruption affects the entire image uniformly and thus benefits less from multi-modal representation.

The pattern is more pronounced on CIFAR100-C, where the increased label complexity amplifies the benefits of our approach. We observe substantial gains on structured corruptions that preserve semantic content while altering appearance (fog: 38.6% vs 39.5%, frost: 29.8% vs 33.8%). The consistent improvements across diverse corruption types—ranging from 0.5% to 3.9%—demonstrate that our multi-cluster architecture provides broad robustness rather than specializing for specific corruption patterns.

An interesting observation is that corruptions appearing later in the sequence (glass, gaussian, pixelate) show larger improvements compared to early corruptions. This suggests that our method's

ability to maintain distinct clusters becomes increasingly valuable as the adaptation history grows, preventing the accumulated bias that affects single-cluster approaches.

Table 5: Classification error rate (%) of the task CIFAR10 $\rightarrow$ CIFAR10-C online continual test-time adaptation evaluated on WideResNet-28 at the largest corruption severity 5. Samples in each corruption are correlatively sampled under the setup PTTA.

| Time | $t \longrightarrow$ | | | | | | | | | | | | | | | |
|---|---|---|---|---|---|---|---|---|---|---|---|---|---|---|---|---|
| Method | motion | snow | fog | shot | defocus | contrast | zoom | brightness | frost | elastic | glass | gaussian | pixelate | jpeg | impulse | Avg. |
| Source | 34.8 | 25.1 | 26.0 | 65.7 | 46.9 | 46.7 | 42.0 | 9.3 | 41.3 | 26.6 | 54.3 | 72.3 | 58.5 | 30.3 | 72.9 | 43.5 |
| BN | 73.2 | 73.4 | 72.7 | 77.2 | 73.7 | 72.5 | 72.9 | 71.0 | 74.1 | 77.7 | 80.0 | 76.9 | 75.5 | 78.3 | 79.0 | 75.2 |
| PL | 73.9 | 75.0 | 75.6 | 81.0 | 79.9 | 80.6 | 82.0 | 83.2 | 85.3 | 87.3 | 88.3 | 87.5 | 87.5 | 87.5 | 88.2 | 82.9 |
| TENT | 74.3 | 77.4 | 80.1 | 86.2 | 86.7 | 87.3 | 87.9 | 87.4 | 88.2 | 89.0 | 89.2 | 89.0 | 88.3 | 89.7 | 89.2 | 86.0 |
| LAME | 29.5 | 19.0 | 20.3 | 65.3 | 42.4 | 43.4 | 36.8 | **5.4** | 37.2 | 18.6 | 51.2 | 73.2 | 57.0 | 22.6 | 71.3 | 39.5 |
| CoTTA | 77.1 | 80.6 | 83.1 | 84.4 | 83.9 | 84.2 | 83.1 | 82.6 | 84.4 | 84.2 | 84.5 | 84.6 | 82.7 | 83.8 | 84.9 | 83.2 |
| NOTE | 18.0 | 22.1 | 20.6 | 35.6 | 26.9 | 13.6 | 26.5 | 17.3 | 27.2 | 37.0 | 48.3 | 38.8 | 42.6 | 41.9 | 49.7 | 31.1 |
| RoTTA | 18.1 | 21.3 | 18.8 | 33.6 | 23.6 | 16.5 | 15.1 | 11.2 | 21.9 | 30.7 | 39.6 | 26.8 | 33.7 | 27.8 | 39.5 | 25.2 |
| ResiTTA | 18.4 | **19.5** | 15.5 | 30.5 | **23.8** | 12.2 | **14.0** | 9.3 | 18.5 | 24.6 | 35.8 | 24.9 | 27.7 | 22.6 | 39.1 | 22.4 |
| **ResiTTA+MCM** | **16.2** | 19.7 | **15.3** | **30.2** | **23.8** | 13.4 | **14.0** | 10.3 | **15.7** | **18.6** | **31.2** | 23.6 | **22.6** | **20.4** | **31.8** | **20.7** |

Table 6: Classification error rate (%) of the task CIFAR100 $\rightarrow$ CIFAR100-C online continual test-time adaptation evaluated on the ResNeXt-29 architecture at the largest corruption severity 5. Samples in each corruption are correlatively sampled under the setup PTTA.

| Time | $t \longrightarrow$ | | | | | | | | | | | | | | | |
|---|---|---|---|---|---|---|---|---|---|---|---|---|---|---|---|---|
| Method | motion | snow | fog | shot | defocus | contrast | zoom | brightness | frost | elastic | glass | gaussian | pixelate | jpeg | impulse | Avg. |
| Source | 30.8 | 39.5 | 50.3 | 68.0 | 29.3 | 55.1 | 28.8 | 29.5 | 45.8 | 37.2 | 54.1 | 73.0 | 74.7 | 41.2 | 39.4 | 46.4 |
| BN | 48.5 | 54.0 | 58.9 | 56.2 | 46.4 | 48.0 | 47.0 | 45.4 | 52.9 | 53.4 | 57.1 | 58.2 | 51.7 | 57.1 | 58.8 | 52.9 |
| PL | 50.6 | 62.1 | 73.9 | 87.8 | 90.8 | 96.0 | 94.8 | 96.4 | 97.4 | 97.2 | 97.4 | 97.4 | 97.3 | 97.4 | 97.4 | 88.9 |
| TENT | 53.3 | 77.6 | 93.0 | 96.5 | 96.7 | 97.5 | 97.1 | 97.5 | 97.3 | 97.2 | 97.1 | 97.7 | 97.6 | 98.0 | 98.3 | 92.8 |
| LAME | **22.4** | **30.4** | 43.9 | 66.3 | **21.3** | 51.7 | **20.6** | **21.8** | 39.8 | **28.0** | 48.7 | 72.8 | 74.6 | **33.1** | **32.3** | 40.5 |
| CoTTA | 49.2 | 52.7 | 56.8 | 53.0 | 48.7 | 51.7 | 49.4 | 48.7 | 52.5 | 52.2 | 54.3 | 54.9 | 49.6 | 53.4 | 56.2 | 52.2 |
| NOTE | 45.7 | 53.0 | 58.2 | 65.6 | 54.2 | 52.0 | 59.8 | 63.5 | 74.8 | 91.8 | 98.1 | 98.3 | 96.8 | 97.0 | 98.2 | 73.8 |
| RoTTA | 31.8 | 36.7 | 40.9 | 42.1 | 30.0 | 33.6 | 27.9 | 25.4 | 32.3 | 34.0 | 38.8 | 38.7 | 31.3 | 38.0 | 42.9 | 35.0 |
| ResiTTA | 29.2 | 33.9 | 39.5 | 39.4 | 28.4 | 29.2 | 26.5 | 24.8 | 33.8 | 33.9 | 37.5 | 38.6 | 31.6 | 37.9 | 41.5 | 33.5 |
| **ResiTTA+MCM** | 27.9 | 32.3 | **38.6** | **37.6** | 26.1 | **27.3** | 24.6 | 23.9 | **29.8** | 32.3 | **36.0** | **37.1** | 29.5 | 36.4 | 39.1 | **31.9** |

# E  HYPERPARAMETER SENSITIVITY ANALYSIS

Table 7 examines sensitivity to maximum clusters $K_{\max}$ and threshold $\tau$, revealing distinct patterns across dataset complexities.

For CIFAR10-C, optimal performance occurs with minimal clustering ($K_{\max} = 1$, $\tau = 0.1$), achieving 20.69% error. Performance degrades monotonically with increased clustering capacity: $K_{\max} = 3$ yields 21.83% (+1.14%), while $K_{\max} = 5$ further deteriorates to 22.82% (+2.13%). This suggests simpler datasets benefit from consolidated memory representations rather than distributed clustering. Similarly, threshold relaxation proves detrimental—increasing $\tau$ from 0.1 to 0.3 raises error to 21.40% (+0.71%), and $\tau = 0.5$ reaches 22.46% (+1.77%), indicating strict similarity criteria are essential for low-complexity scenarios.

CIFAR100-C exhibits contrasting behavior, optimizing at moderate multi-clustering ($K_{\max} = 5$, $\tau = 0.3$) with 31.90% error. Insufficient clusters harm performance ($K_{\max} = 1$: 32.44%, +0.54%), while excessive clustering shows diminishing returns ($K_{\max} = 7$: 32.03%, +0.13%), suggesting an optimal balance between memory diversity and management overhead. The threshold sensitivity differs markedly: while $\tau = 0.3$ performs best, both tighter ($\tau = 0.1$: 32.88%, +0.98%) and looser ($\tau = 0.5$: 32.85%, +0.95%) thresholds degrade performance equally, indicating CIFAR100-C requires balanced clustering criteria—neither too restrictive nor too permissive.

Table 7: Ablation study on hyperparameters for ResiTTA+C2F. We report classification error (%) and relative change $\Delta$ compared to the best setting. Best results are highlighted in **bold**.

| $K_{\max}$ | $\tau$ | Error (%) | $\Delta$ | $K_{\max}$ | $\tau$ | Error (%) | $\Delta$ |
|---|---|---|---|---|---|---|---|
| | | *CIFAR10-C* | | | | *CIFAR100-C* | |
| 1 | 0.1 | **20.69** | – | 1 | 0.3 | 32.44 | +0.54 |
| 3 | 0.1 | 21.83 | +1.14 | 3 | 0.3 | 32.09 | +0.19 |
| 5 | 0.1 | 22.82 | +2.13 | 5 | 0.3 | **31.90** | – |
| – | – | – | – | 7 | 0.3 | 32.03 | +0.13 |
| 1 | 0.1 | **20.69** | – | 5 | 0.1 | 32.88 | +0.98 |
| 1 | 0.3 | 21.40 | +0.71 | 5 | 0.3 | **31.90** | – |
| 1 | 0.5 | 22.46 | +1.77 | 5 | 0.5 | 32.85 | +0.95 |

# F MEMORY SCALING EFFICIENCY COMPARISON

Table 8 compares error rates and runtime performance for baseline configurations (64 samples), naive scaling approach (320 samples), and our proposed MCM configurations across different adaptation methods.

Naive scaling yields minimal accuracy gains with severe runtime overhead across all evaluated methods. RoTTA's memory increase from 64 to 320 samples on CIFAR100-C barely affects error rates ($35.00\% \rightarrow 35.39\%$) but dramatically inflates runtime from 320s to 1556s. PeTTA demonstrates even more problematic scaling behavior, exhausting available memory at 320 samples on CIFAR100-C, while its CIFAR10-C runtime balloons from 572s to 2652s. ResiTTA shows similar inefficiencies, with 320-sample configurations yielding marginal accuracy changes but suffering substantial runtime penalties on both datasets.

In stark contrast, MCM delivers meaningful accuracy improvements while maintaining computational efficiency. ResiTTA+MCM achieves an impressive 20.69% error rate on CIFAR10-C in only 273s—faster than the 359s baseline—and reaches 31.90% on CIFAR100-C in 655s, which is under half the 1492s required for naive scaling with better accuracy results. The pattern holds across other methods: RoTTA+MCM and PeTTA+MCM consistently outperform their naive scaling counterparts in both accuracy and runtime metrics. These results demonstrate that performance gains stem fundamentally from intelligent memory organization rather than simply increasing memory capacity, validating MCM's design philosophy.

Table 8: Memory efficiency comparison under 1-round PTTA. Times are wall-clock for complete round execution. Naive 5× scaling ($64 \rightarrow 320$) increases runtime dramatically without accuracy gains, while MCM achieves lower error with comparable runtime. We use $K_{\max} = 1$ (CIFAR10-C) and $K_{\max} = 5$ (CIFAR100-C).

| Method | CIFAR10−C | | | CIFAR100−C | | |
|---|---|---|---|---|---|---|
| | Cap. | Error (%) | Time (s) | Cap. | Error (%) | Time (s) |
| RoTTA | 64 | 25.20 | 299 | 64 | 35.00 | 320 |
| RoTTA | 320 | 24.81 | 1435 | 320 | 35.39 | 1556 |
| RoTTA + MCM | $64 \times 1$ | **22.59** | **270** | $64 \times 5$ | **33.75** | 501 |
| PeTTA | 64 | 24.30 | 572 | 64 | 35.8 | 688 |
| PeTTA | 320 | 21.70 | 2652 | 320 | - | - |
| PeTTA + MCM | $64 \times 1$ | **21.55** | 881 | $64 \times 5$ | 33.04 | 1043 |
| ResiTTA | 64 | 22.80 | 359 | 64 | 32.50 | 340 |
| ResiTTA | 320 | 23.44 | 1577 | 320 | 32.93 | 1492 |
| ResiTTA + MCM | $64 \times 1$ | **20.69** | **273** | $64 \times 5$ | **31.90** | 655 |

