# OpenReview forum: "A Descriptor-Based Multi-Cluster Memory for Test-Time Adaptation"
_ICLR.cc/2026/Conference — Submitted to ICLR 2026_

### Official Review · Reviewer_74Kv · 2025-10-26

**Soundness:** 3
**Presentation:** 3
**Contribution:** 2
**Rating:** 4
**Confidence:** 4

**Summary:**

This paper introduces Multi-Cluster Memory (MCM), a structured memory mechanism for test-time adaptation (TTA). MCM maintains a dynamic set of clusters, each summarized by lightweight channel-wise mean-variance descriptors. A new sample is assigned to the closest cluster (or spawns a new one), the memory is kept bounded by merging adjacent clusters with the nearest descriptors, and model updates are performed with samples retrieved from the clusters most relevant to the current batch. Extensive experiments on CIFAR-10/100-C, ImageNet-C and DomainNet under the Practical TTA protocol show consistent error reductions when MCM is plugged into three recent memory-based TTA methods; evaluations under the Recurring TTA protocol further demonstrate improved long-term stability.

**Strengths:**

1. MCM yields repeatable accuracy improvements (up to 12.1 % on DomainNet) across multiple strong baselines and datasets, and remains stable over 20 adaptation cycles.
2. Multi-cluster organisation with lightweight descriptors expands coverage of the target distribution while keeping management cost sub-linear, avoiding the linear-growth bottleneck of single-bank approaches.
3. MCM can be integrated into existing TTA frameworks with minimal code changes, making it attractive for practical deployment.
4. Under Recurring TTA, MCM exhibits lower forgetting and smaller variance than its backbones, indicating that the approach mitigates catastrophic drift effectively.

**Weaknesses:**

1. Contrary to the implication in the paper, multi-cluster/prototype organisation has already appeared inside the TTA literature. For example, Contrastive TTA (CVPR 2022) keeps multiple per-sample prototypes for contrastive correction; SoTTA (NeurIPS 2023) maintains several class-level prototypes to combat noisy streams; TEA (CVPR 2024) employs multiple energy centres per class. None of these works. A clear comparative discussion with these TTA-specific precedents is missing.

2. The paper seems simply use the mean and variance of a feature —i.e., a parametric summary of its distribution. The assignment distance is a Euclidean norm between these statistics, which is a simplified (diagonal-covariance) Mahalanobis distance. This connection is neither acknowledged nor compared with earlier works that use similar statistics for domain matching (AdaBN, CORAL, MMD).

3. The current mini-batch may be too small (even batch-size = 1) or internally biased to yield a reliable descriptor. MCM, however, unconditionally assigns that single sample to the nearest cluster (or spawns a new one) based on its lone descriptor. This risks instantaneous mis-clustering: the outlier descriptor can drift an existing centroid or create a new cluster that does not correspond to any true mode of the target distribution.

4. A single mini-batch can itself contain multiple domains (e.g., sunny and rainy frames sampled together). MCM computes one descriptor per sample and immediately assigns each sample to the nearest existing cluster, so samples from different domains may be scattered into separate clusters within the same batch.

5. Insufficient functional validation of MCM's own components. Most of the ablations and experiments measure final classification accuracy and error, but the paper does not include sufficient functional tests that isolate why multi-clustering helps and your functional effectiveness. For example, how the average and maximum pairwise distance of cluster centroids change among your method and other methods, and how this changes through batchs.

6. It seems that there is No cluster-error detection or correction mechanism. If an early outlier drifts the centroid, subsequent samples can be chained into the wrong cluster indefinitely. The paper should clarify whether mis-clustering errors will accumulate and whether there any internal feedback (e.g., entropy rise in a cluster) can trigger some kinds of correction.

7. Many of your experiments are hyper-parameter analysis. However, besides appropriate choice of parameter, results seem to show that the effects are sensitive to hyper-parameters. Performance varies noticeably with τ, K_max and per-cluster capacity N. All values seem to be hand-tuned; I recommend to conduct more small comparison near the best choice of hyper-parameter and among different datasets.

**Questions:**

Q1 Please clarify the conceptual difference between MCM and earlier multi-prototype / multi-exemplar memories What specific design choices make MCM uniquely suited to TTA? Why can multi clustering better describe categories? Can multi clustering really better describe situations that cannot be described by a single cluster.

Q2 Your descriptor is the per-channel mean and variance and the distance is Euclidean. How does this relate to the Mahalanobis distance or to second-order domain-matching objectives such as CORAL or MMD? Have you tried full-covariance distances?

Q3 What is the extra runtime and memory of MCM compared with the single-bank baseline for the same total number of stored samples? Please report some comparison of time and GPU memory. It seems that multiple clusters may need more time and slower than single-bank method.

Q4 I recommend that more recent SOTA method needed to be compared, more 2024 and 2025 method can be provided.

---

> ### Author Response · Authors · 2025-11-23
> **Authors’ Response to Reviewer 74Kv Comments (Weaknesses 1-4)**
>
> Thank you for your thoughtful and constructive feedback. We appreciate the time and effort you have put into reviewing our paper. Below, we address each weakness and question in turn, providing clarifications and describing the corresponding revisions we will make in the updated version.
>
> > **W1** Thank you for the clarification. We agree that multi-centre/prototype ideas have been explored in prior TTA works (as you noted: Contrastive TTA, SoTTA, TEA), and we will revise the wording to avoid implying that “multiple prototypes” are entirely new in TTA.
> >
> > Our contribution instead lies in a different **granularity and role**: we propose a **descriptor-guided Multi-Cluster external memory architecture for PTTA/recurring TTA**, replacing the single unstructured memory pool used by existing memory-based PTTA methods. The memory is dynamically partitioned into clusters to model multi-modal, temporally evolving target streams, with explicit online assignment/maintenance, Adjacent Cluster Consolidation, and relevance-guided retrieval.
> >
> > Specifically, Contrastive TTA uses per-sample prototypes/queues to support a contrastive objective and pseudo-label refinement, rather than structuring an external memory to track multi-domain streams. SoTTA’s multiple class-level centres are mainly for filtering noisy samples, not for multi-modal memory organisation in PTTA. TEA’s multi-centre energy modeling is an in-model parametrization and orthogonal to our external memory management. We will add a concise comparative paragraph in Related Work/Method to make these distinctions explicit.
>
> > **W2** Our descriptor design indeed follows the long-standing practice in TTA literature of using per-channel mean and variance as lightweight statistics for modeling distributional differences. Building on these prior findings, we adopt the same intuition but apply it directly to raw data rather than intermediate features, which avoids dependence on specific backbone architectures and keeps the mechanism broadly compatible. We will clarify these conceptual connections and distinctions to AdaBN/CORAL-style statistics in the revised manuscript.
>
> > **W3** We thank the reviewer for this thoughtful comment. We agree that in the extreme case where the current mini-batch is of size 1 and the sample is an outlier, a noisy descriptor could lead to transient mis-assignment or an unnecessary new cluster (since MCM assigns/spawns clusters based on that descriptor).
> >
> >However, this degenerate setting is not part of the standard Practical/Continual TTA protocols we follow; in common TTA practice the online batches are non-trivial (e.g., batch size 64 in representative baselines), so descriptors are computed from multiple temporally correlated samples and are empirically stable.
> >
> >We acknowledge this limitation in cases where the mini-batch is too small (e.g., batch size = 1) or internally biased to yield a reliable descriptor, and incorporating an explicit outlier-correction or delayed-update mechanism to better handle such deployments would be a valuable direction for future work.
>
> > **W4** Most batches in our setting naturally come from a single domain, so mixed-domain batches are uncommon and typically occur only at transition points. In such rare cases, per-sample assignment is actually preferable: batch-level aggregation could blur sunny and rainy samples into an artificial “cloudy” descriptor, while sample-level assignment preserves domain purity and allows MCM to form clusters that faithfully reflect each distribution. Thus, assigning per sample is the more principled choice.

---

> ### Author Response · Authors · 2025-11-23
> **Authors’ Response to Reviewer 74Kv Comments (Weaknesses 5-7)**
>
> | PeTTA + MCM (ours) | no-ACC | no-RSR |
> |---|---|---|
> | 33.04 | 35.33 | 34.09 |
>
> > **W5** We agree that functional analyses such as centroid-distance evolution would further illustrate *why* multi-clustering helps. While we did not explicitly include centroid-distance plots due to space constraints, the functional behavior of MCM is already implicitly reflected in several experiments. For example, the cluster-growth behavior (Table 7 / Figure 1 of the main paper) and the effectiveness of RSR and ACC in our ablations (above table) already demonstrate that multiple clusters are being meaningfully formed, selected, and consolidated throughout testing—behaviors that cannot arise under a single-cluster design. These results indirectly verify that MCM indeed captures the evolving test-time distribution rather than behaving as a simple enlarged memory buffer. We will incorporate additional visual analyses (e.g., centroid separation over batches) in the updated version for clarity.
>
> > **W6** We agree with the reviewer that MCM does not include an explicit mis-clustering detection/correction module, and we will clarify this limitation in the paper. In practice, error accumulation is mitigated by the memory design: clusters are spawned only when a sample is beyond a distance threshold $\tau$ (otherwise it is absorbed into the nearest cluster), reducing spurious splits; replacement penalizes high-entropy and far-from-centroid samples, so outliers are unlikely to persist and drift centroids; when $K_{\max}$ is reached, ACC merges the closest clusters and keeps the lowest-entropy samples, suppressing drifting/small clusters; and RSR updates the model using only clusters most relevant to the current batch, limiting downstream impact of any accidental mis-cluster. We will also note that adding explicit cluster-error feedback (e.g., entropy monitoring) is a worthwhile future direction.
>
> > **W7** Thank you for the suggestion. We would like to clarify that our main results are \textbf{not hand-tuned per dataset}. Except for CIFAR10-C (where we use a simpler setting with $K_{\max}=1$ due to its lower complexity), we \textbf{fix the same hyper-parameters across all other datasets}: $K_{\max}=5$, Top-$k=5$, and $\tau=0.3$ for CIFAR100-C, ImageNet-C, and DomainNet.
> We agree that performance shows some variation as hyper-parameters change, and we therefore include an additional \textbf{local grid search around the best setting} (shown in the table attached in our response). The results indicate that, although there are fluctuations, the performance remains within a reasonably strong range near the chosen configuration. Nonetheless, we acknowledge that this sensitivity is a limitation of our current design. A more adaptive strategy that dynamically adjusts $\tau$, $K_{\max}$, or per-cluster capacity based on the incoming stream would be more principled, and we view this as a valuable direction for future work.

---

> ### Author Response · Authors · 2025-11-23
> **Authors’ Response to Reviewer 74Kv Comments (Questions)**
>
> Thank you for the reviewer’s questions; we respond point-by-point below.
>
> > **Q1** We thank the reviewer for pointing out these relevant precedents. We clarify that our novelty is **not** the first appearance of multiple prototypes in TTA, but the **first explicit multi-cluster organization of an external PTTA/CTTA memory bank for distribution/mode tracking**.
> >
> > In Contrastive TTA, the “multiple prototypes” are essentially *per-sample exemplars in a feature queue* used for nearest-neighbor voting and contrastive objectives to refine pseudo-labels; they do *not* form a persistent *clustered memory* nor aim to represent temporally evolving domain modes.
> >
> > SoTTA and TEA maintain multiple class-level representatives/centers *inside their adaptation objectives*, mainly for robustness to noisy streams or energy modeling, rather than as a plug-and-play memory structure that can be swapped into existing memory-based PTTA systems.
> >
> > By contrast, our MCM is a *descriptor-based, class-agnostic multi-cluster memory* that (i) groups stored samples into clusters corresponding to different target modes, (ii) spawns/assigns clusters online using lightweight shift descriptors, and (iii) preserves bounded capacity via consolidation, while retrieving only domain-relevant clusters for adaptation.
> >
> > Multi-clustering is beneficial because PTTA/CTTA target streams are *multi-modal and temporally drifting*; a single homogeneous memory cannot distinguish past vs. current modes and thus suffers a staleness–forgetting trade-off. Decomposing memory into clusters yields a more faithful coverage of target modes and empirically improves accuracy.
>
> > **Q2** We agree that our descriptor—per-channel mean and variance—resembles the second-order statistics used in AdaBN, CORAL, and related methods. Our intent was not to introduce a new alignment metric, but to adopt a lightweight, well-validated representation suitable for online Practical TTA. Unlike prior works that use these statistics for domain matching, MCM uses them purely for efficient sample-level clustering and memory organization on raw data, under much stricter latency constraints.
> We have not explored full-covariance descriptors because their computational and memory overhead is difficult to justify in the strict real-time TTA setting. While richer descriptors may further improve robustness, our experiments indicate that even this simple formulation is sufficient for enabling effective multi-cluster construction. We view investigating more expressive yet efficient descriptors as promising future work.
>
> | Method |  | **CIFAR-10-C** |  |  | **CIFAR-100-C** |  |
> |:---|:---:|:---:|:---:|:---:|:---:|:---:|
> |  | Error (%) | Time (s) | GPU (GB) | Error (%) | Time (s) | GPU (GB) |
> | RoTTA (64) | 24.76 | 283  | 6.2  | 35.0 | 242  | 7.9  |
> | RoTTA (320) | 24.81 | 1225 | 16.3 | 34.5 | 1200 | 21.1 |
> | RoTTA + MCM (64×1) | 22.81 | 229  | 4.9  | 35.5 | 207  | 6.3  |
> | RoTTA + MCM (64×5) | 23.30 | 413  | 4.9  | 33.8 | 338  | 6.3  |
> |  |  |  |  |  |  |  |
> | ResiTTA (64) | 22.80 | 313  | 1.4  | 32.5 | 299  | 1.0  |
> | ResiTTA (320) | 23.44 | 1350 | 3.4  | 32.8 | 1241 | 4.3  |
> | ResiTTA + MCM (64×1) | 20.69 | 328  | 1.4  | 32.2 | 308  | 0.7  |
> | ResiTTA + MCM (64×5) | 20.80 | 521  | 1.4  | 31.9 | 444  | 0.7  |
>
> > **Q3** As shown in Fig. 4, we compare runtime for RoTTA, PeTTA, and ResiTTA with and without MCM across different total memory capacities, under matched numbers of stored samples. We also report peak GPU memory usage in the table provided in this response.
> When increasing the total capacity in MCM, we do so by increasing the number of clusters with fixed per-cluster size (64), which leads to only a **modest** runtime increase. In contrast, naively enlarging a single-bank memory incurs **substantial** time and GPU overhead from linearly growing full-bank management/retrieval costs in a larger unstructured bank; importantly, the extra GPU memory is **not** dominated by storing more samples themselves (which are lightweight), but from larger per-step computation and intermediate activations during adaptation and full-bank management.
> This efficiency comes from our design: RSR retrieves and updates only the clusters most relevant to the current batch, so multi-clustering avoids linear per-step cost in total capacity and can be more efficient than single-bank scaling.
>
> > **Q4** Our comparison strategy is specifically focused on methods evaluated under the stringent Practical Test-Time Adaptation (PTTA) protocol, which involves non-i.i.d. sampling.
> We found that most of the new TTA works in 2024/2025 primarily focus on the standard i.i.d. TTA setting. The current selection (including RoTTA, PeTTA, and ResiTTA) represents the most relevant and competitive baselines explicitly designed for the PTTA paradigm.
> By adhering to the established PTTA protocol used by these core works, we ensure a fair and direct comparison against the state-of-the-art in our specific challenging setting.

---

### Official Review · Reviewer_BybS · 2025-10-30

**Soundness:** 3
**Presentation:** 3
**Contribution:** 2
**Rating:** 6
**Confidence:** 4

**Summary:**

This paper introduces *Multi-Cluster Memory (MCM)*, a novel memory management framework for Test-Time Adaptation (TTA). The authors argue that existing memory-based TTA methods, which often use a single, unstructured "single-cluster" memory, suffer from poor *representativeness* (failing to capture the full target distribution) and low *adaptability* (being slow to update under continual shifts).

MCM addresses this by structuring the memory bank into multiple, dynamic clusters. The core components of MCM are:
1.  *Descriptor-Based Management:* It uses lightweight statistical descriptors (channel-wise mean and variance) to characterize samples and define cluster centroids. New samples are assigned to the nearest cluster, or a new cluster is created if the sample is novel (distance > $\tau$).
2.  *Adjacent Cluster Consolidation (ACC):* To maintain a bounded memory capacity ($K \le K_{max}$), MCM merges the *adjacent* cluster pair (in creation sequence) with the closest descriptor distance when the cluster limit is reached.
3.  *Relevance-guided Sample Retrieval (RSR):* For adaptation, the model selects the $N_S$ clusters whose descriptors are most similar to the current mini-batch, ensuring retrieved samples are relevant for the current adaptation step.

The authors demonstrate MCM as a "plug-and-play" module, showing consistent performance improvements when integrated into three existing TTA methods (ROTTA, PeTTA, and ResiTTA). Experiments are conducted on CIFAR-10/100-C, ImageNet-C, and DomainNet under both the *Practical TTA (PTTA)* and *recurring TTA* settings, showing improvement in terms of performance and long-term stability.

**Strengths:**

1. While individual components borrow from prior work, the integration of multi-cluster memory management specifically for TTA is an interesting approach. The combination of descriptor-based clustering, ACC, and RSR within the TTA framework is original
2. The experimental evaluation is comprehensive, covering multiple datasets, baselines, and evaluation protocols (PTTA and recurring TTA). The ablation studies systematically examine hyperparameter sensitivity. The efficiency analysis (Section 4.4) effectively isolates the contribution of memory organization from capacity increases
3. The motivation is well-articulated through Figure 1's visualization of representativeness and adaptability limitations.
4.  MCM demonstrates practical value as a plug-and-play component.

**Weaknesses:**

1. Proposed approach consists of MCM, ACC and RSR. An ablation analysis for the contributions from each of these components is missing.
2. ACC merging heuristics lack theoretical and empirical validation.
3. Key hyperparameter $N_S$ (line 297) for relevant cluster selection is underexplored.
4. Limited novelty, with performance gains mainly on complex datasets.
5. Incomplete comparisons with alternative memory scaling and organization methods.
7. Missing comparison with other clustering baselines.
8. Limited generalizability beyond image classification.
9. The description of ResiTTA (Zhou et al., 2025) as "ResiTTA (Zhou et al., 2025) introduces residual connections to enhance robustness in continual learning for TTA scenarios" is wrong. "Resi" in ResiTTA stands for Resilient, not residual. The method's resilience refers to its ability to maintain rapid adaptation capabilities while mitigating overfitting
10. Categorizing EcoTTA as "Memory-Based" could be misleading. A more accurate classification would be a "Memory-Efficient TTA System".

**Questions:**

1.  **On $K_{max}=1$ for CIFAR10-C:** The finding in Table 7 that $K_{max}=1$ is optimal for CIFAR10-C seems to undermine the paper's core thesis. How do the authors reconcile this? Does this not imply that for some standard benchmarks, the proposed multi-cluster mechanism is actually *harmful* compared to a well-tuned single-cluster baseline (which $K_{max}=1$ effectively is)? Could the authors elaborate on why they believe this happens and what it means for the generality of MCM?
2.  **Justification for Adjacent-Only ACC:** Could the authors provide a stronger justification for limiting ACC to *adjacent* clusters in the creation sequence? Was this choice made purely for computational efficiency (i.e., avoiding a $K^2$ comparison)? An ablation comparing this "adjacent-merge" strategy to a "global-best-pair-merge" strategy (in terms of both accuracy and runtime) would be highly valuable to understand the trade-offs of this design choice.
3.  **Descriptor Robustness:** Given the acknowledged weakness of the current descriptor for noise-based corruptions, have the authors explored alternatives? For example, would using descriptors from a different feature space (e.g., earlier or later in the network) or incorporating other statistics (like histograms or higher-order moments) improve robustness to these shifts without incurring significant computational overhead?
4.  **RSR Sensitivity:** The RSR mechanism selects clusters based on their relevance to the current mini-batch. How sensitive is this process to a highly noisy or anomalous mini-batch? Could a single "bad batch" cause RSR to retrieve unhelpful clusters and trigger negative adaptation?
5. **TRIBE+MCM**: In Table 1, TRIBE without MCM does best for CIFAR-10C. Is it possible to have MCM with TRIBE and have the authors conducted any such experiment?
6. **Sensitivity to hyperparameters**: How are the hyperparameters, such as $K_max$ and threshold $\tau$ for minimum distance to create a new cluster, set in practice? Is it tuned for every dataset, using the test-time adaptation performance? If so, does it mean that the generalizability of the proposed approach is questionable for real-world datasets where we do not have access to labels for the test set? Also, a key hyperparameter $N_S$ (line 297) for relevant cluster selection seems underexplored.\


**Corrections**
1. Fix the subscripts such as: Ci should be C_i in line 260-261; Bt in line 296
2. No space/subscript: Mretrieve (line 298)

---

> ### Author Response · Authors · 2025-11-22
> **Authors’ Response to Reviewer BybS  Comments (Weakness)**
>
> Thank you for your thoughtful and constructive feedback. We appreciate the time and effort you have put into reviewing our paper. Below, we address each weakness and question in turn, providing clarifications and describing the corresponding revisions we will make in the updated version.
>
> | Model Variant | Error Rate (%) |
> |-|-|
> |Full model|33.04|
> |w/o ACC|35.33|
> |w/o RAR|34.09|
>
> > **W1** We have included ablation studies for the two proposed management mechanisms, ACC and RSR. As shown in the above table, removing either component leads to a noticeable performance degradation, while employing both simultaneously yields the best results. This demonstrates that the two mechanisms are complementary and jointly contribute to the effectiveness of our method.
>
> > **W2** The primary motivation behind ACC is to prevent the number of clusters in memory from growing without bound. To preserve both a meaningful memory distribution and memory efficiency, we adopt an approach inspired by the video‐consolidation mechanism used in MovieChat, where the most similar adjacent clusters are merged. This design provides the desired functionality of keeping the memory compact while retaining representative clusters. Our experiments in W1 further validate that ACC is an essential component of MCM, as removing it leads to noticeable performance degradation.
>
> | $K_{max}$ | $N_S$| Error Rate (%) |
> |-----------:|------:|---------------:|
> | 6 | 5 | 33.42 |
> | 6 | 6 | 33.58 |
> | 5 | 4 | 33.63 |
> | 5 | 5 | 33.75 |
> | 6 | 4 | 33.82 |
> | 4 | 4 | 33.86 |
>
> > **W3** Above we include an additional experiment analyzing the effect of varying $N_S$. The results show that RSR generally benefits from selecting a number of clusters close to $K_{max}$. This suggests that, most of the time, all clusters remain informative and contribute meaningfully during adaptation. When a domain shift occurs, only clusters representing the previous domain become less relevant—an effect that can also be observed qualitatively in Figure 1 of the main paper.
>
> > **W4** We believe that achieving larger gains on complex datasets is not a weakness but rather a strength of our method. Complex benchmarks naturally exhibit richer and more heterogeneous distribution shifts, making them the most challenging—and also the most meaningful—test cases for TTA. The fact that MCM consistently improves performance in these scenarios highlights its practical value. Regarding novelty, our contribution focuses on demonstrating that the distribution of stored memory samples in TTA can be further expanded and better aligned with the evolving test-time distribution. The proposed mechanisms are designed precisely to manage and refine this memory distribution, enabling more faithful and adaptive utilization of past samples during test-time adaptation.
>
> > **W5** To the best of our knowledge, existing memory-based TTA methods do not explicitly focus on scaling strategies for enlarging or structuring the memory bank. Regarding memory organization, we have already included the most relevant approaches as baselines in our comparisons. If there are additional references that the reviewer believes are directly related to memory scaling or organization in TTA, we would greatly appreciate further pointers so that we can include them in our discussion.
>
> > **W6** While there do exist several clustering-based approaches, their methodological focus is quite different from ours. These works primarily perform clustering in feature or prediction space, rather than clustering memory samples as we do. Moreover, most of them are not designed for test-time adaptation (TTA), and to our knowledge, virtually none of them operate under the Practical TTA setting, which imposes stricter constraints on memory, data streaming, and recurring domain shifts. Thus, they are not directly comparable baselines for our problem formulation.
>
> > **W7** We acknowledge this limitation: our method is evaluated only on image classification. Its generalizability to other tasks/modalities (e.g., detection, segmentation) is not yet verified and will be explored in future work.
>
> > **W8**&**W9** Thank you for pointing this out. We acknowledge both issues and will correct them in the updated version.

---

> > ### Author Response · Authors · 2025-11-22
> > **Authors’ Response to Reviewer BybS Comments (Questions)**
> >
> > Thank you for the reviewer’s questions; we respond point-by-point below.
> >
> > >**Q1** We agree that the CIFAR10-C results indicate that using five clusters is not universally optimal across all datasets. This behavior is expected, as the optimal number of clusters naturally depends on the complexity of the test-time distribution. CIFAR10-C exhibits relatively simple and homogeneous shifts, making a single cluster sufficient and even preferable in this case. Rather than undermining MCM, this observation highlights that cluster cardinality should ideally be adapted to the dataset’s distributional characteristics. Exploring mechanisms that dynamically adjust the number of clusters during test-time adaptation is therefore an important direction for future work.
> >
> > > **Q2** Yes, our choice of adjacent-only consolidation is primarily motivated by efficiency. Moreover, since clusters are created sequentially during test-time streaming, adjacent clusters typically correspond to temporally proximate segments of the evolving target distribution. In such settings, adjacent clusters are generally more likely to share similar distributional characteristics, making adjacency a reasonable and effective heuristic for consolidation while avoiding the computational overhead of global pairwise comparisons.
> >
> > > **Q3** We acknowledge the reviewer’s observation regarding the sensitivity of our mean–variance descriptor to noise-based corruptions. This is indeed a limitation of adopting such a lightweight statistic. In the current work, we did not further explore alternative descriptors, as our design goal was to keep MCM computationally minimal to satisfy the strict efficiency requirements of the Practical TTA setting. Extracting descriptors from intermediate network features or incorporating richer statistics (e.g., histograms or higher-order moments) would likely offer improved robustness, but may introduce additional computational overhead that needs to be carefully examined. Exploring such descriptor variants—while preserving the lightweight nature of MCM—is an interesting and valuable direction that we intend to investigate in future work.
> >
> > > **Q4** While it is true that an anomalous mini-batch may temporarily lead RSR to select suboptimal clusters, this effect is naturally self-correcting within MCM. If the incoming samples reflect a genuinely shifted distribution, MCM will rapidly allocate a new cluster and begin accumulating samples there, enabling the system to adapt quickly to the new domain. Conversely, if the batch is only a short-term outlier, its influence remains limited because RSR will revert to selecting the more representative clusters as subsequent batches arrive. In practice, such transient deviations do not meaningfully destabilize the adaptation process.
> >
> > > **Q5** TRIBE’s mechanism relies on maintaining class-wise statistics within the BatchNorm layers, whereas MCM retains sample-level memory banks. These two forms of memory operate at fundamentally different levels and are not directly compatible. Moreover, TRIBE performs class balancing within the BN layers, while MCM, following RoTTA and PeTTA, already enforces class balance at the sample-bank level. Finally, across the more challenging datasets in Table 1, the baseline methods equipped with MCM already outperform TRIBE, indicating that MCM provides stronger adaptation benefits under complex distribution shifts.
> >
> > **Local grid search results of RoTTA+MCM on CIFAR-100-C**
> > | Error Rate (%) | Clusters $K$ | Top-$k$ | Threshold $\tau$ |
> > |---:|---:|---:|---:|
> > | 33.42 | 6 | 5 | 0.3 |
> > | 33.58 | 6 | 6 | 0.3 |
> > | 33.63 | 5 | 4 | 0.3 |
> > | 33.75 | 5 | 5 | 0.3 |
> > | 33.82 | 6 | 4 | 0.3 |
> > | 33.86 | 4 | 4 | 0.3 |
> > | 34.03 | 6 | 6 | 0.4 |
> > | 34.04 | 4 | 4 | 0.4 |
> > | 34.06 | 5 | 4 | 0.4 |
> > | 34.18 | 5 | 5 | 0.4 |
> > | 34.36 | 5 | 4 | 0.5 |
> > | 34.38 | 6 | 4 | 0.4 |
> > | 34.54 | 6 | 5 | 0.4 |
> > | 34.78 | 6 | 6 | 0.5 |
> > | 34.81 | 4 | 4 | 0.5 |
> > | 34.85 | 6 | 5 | 0.5 |
> > | 34.94 | 5 | 5 | 0.5 |
> > | 34.95 | 6 | 4 | 0.5 |
> >
> >
> > > **Q6** We do **not** tune hyperparameters per dataset at test time, nor use any test labels. We pick a **single stable default** from a coarse local grid on a reference setting, then **fix it for all datasets/corruptions**. The local grid results below show only mild variation within a reasonable range, indicating low sensitivity. Importantly, except for CIFAR-10C where we use a **single-cluster setting $K=1$** (as stated in the main text), we adopt the **same** $(K=5,\ \text{top-}k=5,\ \tau=0.3)$ for the other three more complex, multi-class datasets (CIFAR-100-C, ImageNet(-C), and DomainNet) **without special tuning**. The top-$k$ parameter is explored jointly with $K$ and $\tau$ (with $\text{top-}k \le K$ by design). We will clarify this protocol in the revision.

---

### Official Review · Reviewer_2mCb · 2025-10-31

**Soundness:** 2
**Presentation:** 2
**Contribution:** 2
**Rating:** 2
**Confidence:** 3

**Summary:**

The paper addresses the lack of representativity and adaptability in memory-based test-time adaptation (TTA) methods. It introduces Multi-Cluster-Memory (MCM), a memory mechanism that maintains multiple clusters of previously seen samples instead of a single global buffer. Each cluster stores samples along with their statistical descriptors, and new samples are either assigned to the nearest existing cluster or used to start a new one. When the maximum number of clusters (Kmax) is reached, an Adjacent Cluster Consolidation (ACC) step merges the most similar clusters to free space. During adaptation, the model retrieves samples from the most relevant clusters through a Relevant Sample Retrieval (RSR) step, based on proximity to the current batch. Experiments on CIFAR-10-C, CIFAR-100-C, ImageNet-C, and DomainNet show that MCM improves over standard single-cluster memory-based baselines.

**Strengths:**

* The technical proposal of maintaining multiple evolving clusters for online adaptation is conceptually interesting.
* Results are presented on multiple datasets, including both corruption-based (CIFAR-C, ImageNet-C) and domain shift benchmarks (DomainNet).

**Weaknesses:**

* The paper is difficult to follow, especially in describing the TTA setups and the details of method.

* The ablation shows that gains from using multiple clusters are modest (less than 2 % improvement between Kmax = 1 and 5). The evidence is insufficient to justify the added complexity.

* The authors criticize sequential updates (L80, L304), but it’s not clear how MCM avoids updating samples sequentially, given that samples arrive sequentially.

* Large variability in error reductions (Table 1) — across methods, datasets, and even within the same method or dataset.

**Questions:**

* What exactly is Practical TTA? Does it refer to the case of non-IID test samples? If yes, what is the setup to realize this?

* In Eq. 1, what do the student model fₛ and teacher model fₜ represent in the context of the TTA setup?

* In what sense is MCM cost-efficient compared to the single-cluster memory approach?

* Could you confirm that when the cluster limit is reached, only two clusters are merged, freeing a single slot so that the number of clusters becomes Kmax − 1?

* Is Kmax = 1 equivalent to the single-cluster memory baseline?

* What is the retrieval mechanism for the single-cluster case, and how does MCM + RSR compare directly to Single-Cluster + RSR?

* What factors explain the high intra-dataset or intra-method variability in the performance gains reported in Table 1?

---

> ### Author Response · Authors · 2025-11-21
> **Authors’ Response to Reviewer Comments (Weaknesses)**
>
> Thank you for your thoughtful and constructive feedback. We appreciate the time and effort you have put into reviewing our paper. Below, we address each weakness and question in turn, providing clarifications and describing the corresponding revisions we will make in the updated version.
>
> ---
>
> > **W1** Below, we provide a concise clarification of the TTA setting and the core design principles underlying our method.
>
> **TTA setups**
>
> - **TTA**: Test-time adaptation (TTA) updates a pretrained model at inference time using only unlabeled test data so it can adjust to distribution shifts between the training and testing data on the fly.
> - **CTTA**: Continual TTA considers a test data stream with multiple domain switches, where the model must keep adapting online as the distribution changes.
> - **PTTA**: Practical TTA is developed from CTTA by re-sampling the stream to make it explicitly non-i.i.d., while jointly modeling realistic distribution changes and temporal correlations in the incoming data.
>
> **Method**
>
> Single-cluster memories put all target samples into one shared pool without separating different shift patterns, so they may not represent the full variety of target data and their adaptability to new target distributions under continual shift can be restricted.
>
> Therefore, we adopt a multi-cluster memory to better represent diverse target shifts. Since test-time data arrive unlabeled and their domain identity is unknown, we use lightweight shift descriptors to assign each sample to the closest cluster or to spawn a new one. To avoid unbounded cluster growth (which would increase computation) and to prevent forgetting that could arise from a naive FIFO policy, we introduce Adjacent Cluster Consolidation (ACC), which merges the two most similar clusters based on descriptor distance. This keeps total memory bounded while preserving cluster diversity. At test time, Relevance-guided Sample Retrieval (RSR) selects samples from clusters most consistent with the current distribution for adaptation, achieving an efficient balance among effectiveness, diversity, and cost.
>
> Our approach builds on Multi-Cluster Memory (MCM) to capture target-domain diversity, with three coupled components (Fig. 2):
> - **Descriptor-based Clustering**: group samples into clusters via lightweight mean/variance descriptors.
> - **Adjacent Cluster Consolidation (ACC)**: merge the two closest clusters when memory is full to keep capacity bounded.
> - **Relevance-guided Sample Retrieval (RSR)**: retrieve the most relevant cluster(s) using current test-batch statistics for adaptation.
>
>
> ---
>
> > **W2** As shown in Table 1, our method yields consistent improvements across multiple baselines and datasets, with several settings exceeding a 2% gain. While multiple clusters introduce some management overhead, this cost is substantially lower than enlarging a single-cluster memory, as evidenced in Fig. 4. We agree that there is room to further optimize this trade-off, and designing an even more efficient multi-cluster strategy is an important direction for future work.
>
> ---
>
> > **W3** The intuition is straightforward: a single-cluster memory must gradually replace individual samples before it can represent a new domain, so adaptation happens only after sufficient overwriting. In contrast, MCM can directly allocate a new cluster for incoming samples, allowing the new domain to be represented immediately without overwriting existing information.
>
> ---
>
> > **W4** While the magnitude of improvement varies across methods and datasets, the overall trend in Table 1 consistently shows performance gains. Such variability is expected because different architectures and corruption types respond differently to test-time adaptation. The key point is that MCM provides stable improvements across all evaluated settings, even though the exact reduction cannot be identical for every combination.

---

> ### Author Response · Authors · 2025-11-21
> **Authors’ Response to Reviewer Comments (Questions)**
>
> Thank you for the reviewer’s questions; we respond point-by-point below.
>
>
> > **Q1 (Reviewer):** What exactly is Practical TTA? Does it refer to the case of non-IID test samples? If yes, what is the setup to realize this?
>
> **Response:** Thank you for the question. In our paper, *Practical Test-Time Adaptation (PTTA)* follows the definition introduced in RoTTA. Conceptually, PTTA denotes a deployment-oriented setting where:
> 1. the model sees a **single streaming pass of test samples** drawn from multiple target domains/corruptions,
> 2. the test stream is **non-i.i.d. and temporally correlated**, and
> 3. adaptation is performed **online using only unlabeled test data**, without any access to the source training set.
>
> So yes, PTTA explicitly includes the case of non-i.i.d. test samples. As detailed in Section 4.1, the non-i.i.d. setup is realized by:
> 1. **Correlative sampling using a Dirichlet distribution:** we use a Dirichlet process to simulate temporal correlation between successive test batches.
> 2. **Standard parameter $\delta=0.1$:** we set the Dirichlet parameter to $\delta=0.1$ in all experiments.
>
> ---
>
> > **Q2 (Reviewer):** In Eq. (1), what do the student model $f_s$ and teacher model $f_t$ represent in the context of the TTA setup?
>
> **Response:** In Eq. (1), $f_s$ and $f_t$ form a standard Mean-Teacher–style student–teacher pair commonly used in test-time adaptation. They share the same architecture and are initialized from the same source-pretrained model. The student $f_s$ is directly updated during TTA by minimizing the consistency loss on memory samples, while the teacher $f_t$ provides stable targets (pseudo-labels) and is updated only via an exponential moving average (EMA) of the student’s parameters rather than backpropagation. This setup is widely used in prior TTA methods such as CoTTA and RoTTA, as well as in semi-supervised learning under the Mean Teacher paradigm.
>
> > **Q3 (Reviewer):** In what sense is MCM cost-efficient compared to the single-cluster memory approach?
>
> **Response:** MCM avoids comparing each incoming sample with the entire memory. A single-cluster memory must score and compare against all stored samples, whereas MCM first matches lightweight descriptors over a few clusters to find the relevant one, then compares only within that cluster. This hierarchical filtering reduces management overhead under the same memory size.
>
> ---
>
> > **Q4 (Reviewer):** Could you confirm that when the cluster limit is reached, only two clusters are merged, freeing a single slot so that the number of clusters becomes $K_{\max}-1$?
>
> **Response:** We merge exactly two clusters into one, reducing the count by one per consolidation. But consolidation happens only after a new cluster is created and the limit is briefly exceeded, so the count goes from $K_{\max}+1$ back to $K_{\max}$ (not to $K_{\max}-1$). Thus we always keep at most $K_{\max}$ clusters.
>
> ---
>
> > **Q5 (Reviewer):** Is $K_{\max}=1$ equivalent to the single-cluster memory baseline?
>
> **Response:** No. With $K_{\max}=1$ we still use a cluster descriptor in memory management, and the replacement score $H(x)$ includes an additional descriptor-distance term. So it is not the same as the original single-cluster baseline.
>
>
> ---
>
> > **Q6 (Reviewer):** What is the retrieval mechanism for the single-cluster case, and how does MCM + RSR compare directly to Single-Cluster + RSR?
>
> **Response:** In the single-cluster case, retrieval samples adaptation data from the entire memory bank without any cluster-level selection, as in RoTTA, PeTTA, and ResiTTA. In contrast, MCM+RSR computes a relevance score for each cluster (via descriptor distance to the current mini-batch) and samples only from the $N_S$ most relevant clusters. If RSR is applied to a single-cluster memory, there is only one cluster to select, so it degenerates to sampling from the whole memory and is effectively identical to the original single-cluster baseline. Thus, the difference comes solely from cluster-level selection, which exists in MCM+RSR but not in single-cluster memory.
>
> > **Q7 (Reviewer):** What factors explain the high intra-dataset or intra-method variability in the performance gains reported in Table 1?
>
> **Response:** The gain variability mainly stems from how our memory design interacts with each dataset and baseline. Datasets differ in shift complexity (e.g., number of modes, corruption strength/structure, baseline robustness), so the headroom for improvement from a more expressive memory varies: heterogeneous or multi-modal shifts benefit more, while simpler shifts leave less room. Baselines also differ in how much they rely on stored samples and how aggressively they update memory, so memory-sensitive methods gain more from multi-cluster memory, whereas others see smaller but consistent gains. Overall, this variability reflects different degrees of “memory-limitedness” across settings, not instability of our approach.

---

> > ### Comment · Reviewer_2mCb · 2025-11-25
> >
> > I thank the authors for the clarifications. Are the authors aware of recent advancements in TTA, particularly layer-selection–based approaches such as GALA [1] and PALM [2]? It would be helpful to understand how the proposed cluster-management strategy compares to these methods.
> >
> > Additionally, is there any statistically significant evidence that justifies having the more complex setting with $K_{\max}$ > 1 rather than the simple $K_{\max}$=1? As presented, the current ablation does not provide sufficient motivation for this.
> >
> > [1] Sahoo, Sabyasachi, et al. _A layer selection approach to test-time adaptation._ AAAI 2025.
> >
> > [2] Maharana, Sarthak Kumar, Baoming Zhang, and Yunhui Guo. _PALM: Pushing adaptive learning rate mechanisms for continual test-time adaptation._ AAAI 2025.

---

> ### Author Response · Authors · 2025-11-26
> **Official Comments by Authors**
>
> **First**, regarding the two recent advancements you mentioned, we are indeed aware of both works. However, these methods are specifically designed for the Continual TTA setting, whereas our work focuses on a more general and practical TTA scenario. For this reason, we did not include them in the comparison.
>
> That said, their elegant layer-selection–based strategies are largely orthogonal to our approach and can be integrated into our framework. In the revised version, we will include a brief comparison and also discuss how such techniques may be incorporated into our TTA system in future extensions.
>
> **Second**, Regarding the reviewer’s question on whether using $K_{max}>1$ in MCM is necessary: this choice fundamentally depends on the complexity of the target distribution. For simpler datasets such as CIFAR-10, the distribution is nearly unimodal, and our results indeed show that $K_{max}=1$ performs the best.
>
> In contrast, for more complex datasets such as CIFAR-100, ImageNet-C, and DomainNet, our experiments consistently demonstrate that $K_{max}>1$ yields clear and stable performance gains, as presented in the paper.
>
> More importantly, in realistic deployment scenarios of practical TTA, the underlying data distributions are typically far more complex than those in standard benchmark datasets. Under such conditions, a multi-cluster memory design with $K_{max}>1$ becomes not only beneficial but arguably necessary for capturing diverse domain shifts.

---

> > ### Comment · Reviewer_2mCb · 2025-11-28
> >
> > * You state that the compared methods were designed for a continual TTA setting, yet you do include CoTTA, which is a typical continual TTA approach. If your method is intended to work in a more general and practical TTA scenario, then it should also be demonstrated to work in the less general and "easier" continual TTA case.
> >
> > * Being orthogonal should be a motivation to show how these methods integrate with the methods, not a reason for excluding them. Some form of comparison is necessary to appreciate the progress that your method brings.
> >
> > * I agree that domain shifts are diverse. However, this alone does not validate the fundamental multi-cluster assumption. A more rigorous validation would require statistically significant evidence on more than one dataset. As far as I can see, this is not established in the current results.

---

> > > ### Author Response · Authors · 2025-12-03
> > > **Official Comments by Authors**
> > >
> > > * There may be a slight misunderstanding here. We have already integrated the multi-cluster technique into three different baselines and evaluated it across four datasets, all showing significant improvements. It is not limited to a single dataset as you suggested.
> > >
> > > * Regarding the baselines you mentioned under the newer CTTA setting, your point is correct—CTTA is indeed an easier setting, and our method can certainly be integrated with those baselines. Our initial intention was to focus on PTTA, as we believed it was sufficient to demonstrate the effectiveness of our method. If you feel this is still insufficient, we are willing to include additional experiments. However, due to the current time constraints, we can provide them in a subsequent updated version to more comprehensively validate the effectiveness of MCM.

---

### Official Review · Reviewer_6FTR · 2025-10-31

**Soundness:** 3
**Presentation:** 3
**Contribution:** 2
**Rating:** 4
**Confidence:** 3

**Summary:**

The paper proposes Multi-Cluster Memory (MCM) for better long-term adaptation under continual and partial test-time adaptation. Instead of keeping a single-cluster memory of confident test samples, MCM organizes them into clusters using channel-wise statistical descriptors and assigns each new sample to the nearest cluster or creates a new one. To bound memory, it introduces Adjacent Cluster Consolidation (ACC) that merges the most similar adjacent cluster pair, retains the N lowest-uncertainty samples and evolves over time with domain changing. For adaptation, Relevance-guided Sample Retrieval (RSR) selects clusters most similar to the current mini-batch and trains with the standard Mean-Teacher consistency objective. Experiments on CIFAR-10/100-C, ImageNet-C, and DomainNet under PTTA show consistent improvements over memory-based baselines (RoTTA, PeTTA, ResiTTA).

**Strengths:**

- **Targeted retrieval designed for PTTA.** RSR ties the adaptation batch to clusters most similar to the current stream, which is conceptually aligned with continually changing-domain  shifts and avoids mixing unrelated modes during self-training.
- **Modular idea; simple but effective mechanics.** Using per-channel mean/variance descriptors to structure memory is compute-light and easy to integrate, which is a good fit with TTA and easy to integrate with current memory-based methods.
- **Consistent gains across backbones and baselines.** The recurring-shift evaluation also shows stability under long time adaptation.

**Weaknesses:**

- **Novelty is incremental relative to prior "memory-based TTA".** The main contribution is structuring the memory with simple descriptors and adding an ACC/RSR policy to replace the standard single memory. While practical, the core mechanics for memory design (distance-based assignment, merging, uncertainty-aware replacement) resemble standard online clustering and memory curation ideas; stronger positioning versus alternative memory-bank designs would help.
- **Scope of descriptors.** The method fixes channel-stat descriptors; it’s unclear how robust MCM is to the choice of layer or to alternative descriptors (e.g., low-dimensional embeddings or BN running stats). Also I am curious if Transformer-based model can gain from MCM design.
- **Limited component ablations** It seems like there lacks a clean ablation that removes ACC or RSR individually (beyond hyperparameter sweeps). A table with “no-ACC / no-RSR / both” would clarify where the gains come from.
- **Limited visualization and interpretability.** The paper only provides one qualitative illustration (Fig. 1) to motivate representativeness vs. adaptability. More visual analyses—e.g., memory distributions or cluster evolution plots across different datasets and stages—would make the benefits of MCM over single-memory schemes more tangible and convincing.
- **Sensitivity to Hyperparameters.** Although MCM is described as a plug-and-play module, its performance depends on several hyperparameters.

**Questions:**

1. It would be helpful if the paper could provide quantitative results on runtime and memory usage—such as end-to-end throughput  and peak GPU memory—for MCM compared to a single-cluster memory of equivalent total capacity.
2. In Table 1, TRIBE performs surprisingly well, in some cases matching or even surpassing MCM. Could the authors elaborate on possible reasons for this—e.g., differences in adaptation schedule, update frequency, or experimental setting?

---

> ### Author Response · Authors · 2025-11-22
> **Authors’ Response to Reviewer 6FTR Comments (Weaknesses)**
>
> Thank you for your thoughtful and constructive feedback. We appreciate the time and effort you have put into reviewing our paper. Below, we address each weakness and question in turn, providing clarifications and describing the corresponding revisions we will make in the updated version.
>
> > **W1.** While prior memory-based TTA methods indeed incorporate assignment and uncertainty-aware replacement, these approaches inherently operate under a **single-cluster memory** assumption, which limits their ability to scale and often leads to slow adaptation when the test-time distribution shifts. Our contribution is to highlight this limitation and introduce a **multi-cluster memory (MCM)** formulation that can expand efficiently as new distributions appear. The presence of multiple clusters naturally gives rise to the need for a **consolidation (merging) mechanism**, which does not exist in single-cluster memories. Thus, the novelty of MCM lies not in reusing standard memory components, but in enabling a scalable, distribution-adaptive memory structure that better accommodates domain transitions.
>
> > **W2.** We apologize for the inaccurate statement in the paper. The descriptor computation is actually based on the raw data (i.e., the image samples), rather than features extracted by a CNN. We will correct this in the revised version. Furthermore, relying on raw data makes our mechanism naturally compatible with transformer-based models as well.
>
> | Model Variant | Error Rate (%) |
> |-|-|
> |Full model|33.04|
> |w/o ACC|35.33|
> |w/o RAR|34.09|
>
> > **W3.** We have included ablation studies for the two proposed management mechanisms, ACC and RSR. As shown in the above table, removing either component leads to a noticeable performance degradation, while employing both simultaneously yields the best results. This demonstrates that the two mechanisms are complementary and jointly contribute to the effectiveness of our method.
>
> > **W4.** Additional visual analyses—particularly those involving cross-dataset settings—will be included in the updated version of the paper.
>
> > **W5.**
> For the four datasets evaluated in Table 1 of the main paper, we used a single shared set of hyperparameters for CIFAR100-C, ImageNet-C, and DomainNet, as this configuration consistently provided the best performance across these benchmarks. Only CIFAR10-C required fewer clusters to achieve better results. Our understanding is that this is because CIFAR10-C contains significantly fewer classes, resulting in a simpler test-time sample distribution that does not benefit from having many clusters to represent it. We acknowledge that our architecture introduces several hyperparameters, as we aim to present a more general framework for PTTA. Providing dynamic control over memory management and clustering behaviors is inherently a more flexible and versatile design choice, resulting in additional hyperparameters during optimization. We appreciate the comment and view this challenge as an area for future investigation to develop adaptive mechanisms that automatically weight these components.

---

> ### Author Response · Authors · 2025-11-22
> **Authors’ Response to Reviewer 6FTR Comments (Questions)**
>
> | Method                     | Error Rate ⬇️ | Latency for 1 run TTA (s) ⬇️ | Peak GPU Usage (GB) ⬇️ |
> |----------------------------|---------------|-------------------------------|-------------------------|
> | RoTTA (64 samples in mem)  | 25.20         | 299                           | 6.2                     |
> | RoTTA (320 samples in mem) | 24.81         | 1435                          | 16.3                    |
> | RoTTA + MCM (64×1)         | 22.59         | 229                           | 6.3                     |
> | RoTTA + MCM (64×5)         | 23.30         | 413                           | 6.3                     |
>
> >**Q1** We added runtime and memory comparisons between MCM and a single-cluster memory of equal total size. Consistent with the trends shown in Figure 4 of the main paper, MCM demonstrates clear efficiency and accuracy improvements as the number of stored samples scales. Notably, even as the memory size increases, the number of replay samples selected via RSR remains unchanged, keeping the peak GPU usage stable.
>
> >**Q2** TRIBE’s mechanism relies on maintaining class-wise statistics within the BatchNorm layers, whereas MCM retains sample-level memory banks. These two forms of memory operate at fundamentally different levels and are not directly compatible. Moreover, TRIBE performs class balancing within the BN layers, while MCM, following RoTTA and PeTTA, already enforces class balance at the sample-bank level. Finally, across the more challenging datasets in Table 1, the baseline methods equipped with MCM already outperform TRIBE, indicating that MCM provides stronger adaptation benefits under complex distribution shifts.

---

### Comment · Area_Chair_xcst · 2025-11-27
**Reminder: Engage in Discussions and Finalize Your Review**

Dear Reviewers,

Thank you for your valuable reviews. With the Reviewer-Author Discussions deadline approaching, please take a moment to read the authors’ rebuttal and the other reviewers’ feedback, and participate in the discussions and respond to the authors. Finally, be sure to complete the “Final Justification” text box and update your “Rating” as needed. Your contribution is greatly appreciated. I will flag irresponsible (final) reviews and/or any reviewers not participating in discussions.

Reviewers are expected to stay engaged in discussions, initiate them, respond to authors’ rebuttal, ask questions, and listen to answers to help clarify remaining issues.

It is not OK to stay quiet.

It is not OK to leave discussions till the last moment.

If authors have resolved your (rebuttal) questions, do tell them so.

If authors have not resolved your (rebuttal) questions, do tell them so too.

Thanks,

AC

---

### Meta-Review · Area_Chair_xeJP · 2025-12-23

**Summary:**

The reviewers’ initial scores range from 4 / 2 / 6 / 4, and collectively raise significant concerns regarding the motivation, empirical effectiveness, and validity of the multi-cluster assumption underlying the proposed Multi-Cluster Memory (MCM).
More specifically, multiple reviewers question (i) whether the proposed descriptor design is sufficiently justified, (ii) under what conditions the multi-cluster memory provides meaningful advantages over a single-cluster design, and (iii) whether such advantages are consistently observed across datasets and baselines.
In addition, the lack of component-level ablations and informative visual analyses makes it difficult to determine whether the reported gains stem from the multi-cluster design itself.

From the AC’s perspective, while some clarifications are provided in the rebuttal, the current evidence does not sufficiently demonstrate that the multi-cluster design is broadly effective. Moreover, a serious inconsistency is observed between the paper and the rebuttal regarding whether the descriptor statistics are computed from feature maps or from raw inputs, which further undermines the clarity and reliability of the proposed method.

Taken together with the initial score distribution, the recommended decision for this paper is reject.

**Reviewer Concerns:**

**(W2 of Reviewer 6FTR, W2 of Reviewer 74Kv)** The concern remains unresolved.
While the authors state that the descriptor is computed from raw data (image samples), this directly contradicts Section 3.3, which explicitly defines it as statistics computed from "feature maps" (Eq. 2). More importantly, the rebuttal does not address the core questions regarding the justification and robustness of the descriptor design (e.g., choice of channel-wise statistics and layer sensitivity). The applicability to Transformer architectures also remains theoretically unjustified beyond the vague claim of input-level compatibility.

**(W3 of Reviewer 6FTR, W1 of Reviewer BybS, and W5 of Reviewer 74Kv)** Component-level ablation studies are important to clearly attribute the gains of the proposed method. While the rebuttal presents ACC/RSR ablations, the experimental setup is unclear and the results are not reflected in the revised manuscript.

**(W4 of Reviewer 6FTR)** Additional visual analyses could strengthen the motivation and demonstrate the effectiveness of MCM across datasets, but such visualizations are not incorporated in the revised manuscript.

**(W2 and W4 of Reviewer 2mCb)** These are critical. The proposed multi-cluster design does not demonstrate consistent benefits across datasets, which weakens the claim of its general effectiveness. In particular, the largest reported gain (12.13% for ResiTTA on DomainNet in Table 1) relies on a self-implemented baseline (54.76%) that appears substantially weaker than established results, while other baselines seem to be directly taken from prior work rather than consistently reproduced (Line 273). To more clearly demonstrate the efficacy of MCM, the authors should have reported performance gains relative to consistently reproduced baselines.

**(W7 of Reviewer BybS)**
While the paper motivates TTA with autonomous driving and robotic manipulation examples (Lines 34–41), the experiments are restricted to image classification, which limits the strength of the broader motivation.

**Reviewer Scores:**

Based on the unresolved concerns outlined in the Reviewer Concerns section, it is unlikely that the reviewers would have increased their scores after full discussion, as the rebuttal does not fully address the core issues raised.

---

### Decision · Program_Chairs · 2026-01-26

Reject